# Greedy Approximation Algorithms for Active Sequential Hypothesis Testing

**Kyra Gan**\*, **Su Jia**,\* **Andrew A. Li**
Carnegie Mellon University
Pittsburgh, PA 15213
{kyragan,sujia,aali1}@cmu.edu

## Abstract

In the problem of *active sequential hypothesis testing* (ASHT), a learner seeks to identify the *true* hypothesis from among a known set of hypotheses. The learner is given a set of actions and knows the random distribution of the outcome of any action under any true hypothesis. Given a target error $\delta > 0$, the goal is to sequentially select the fewest number of actions so as to identify the true hypothesis with probability at least $1 - \delta$. Motivated by applications in which the number of hypotheses or actions is massive (e.g., genomics-based cancer detection), we propose efficient (greedy, in fact) algorithms and provide the first approximation guarantees for ASHT, under two types of adaptivity. Both of our guarantees are independent of the number of actions and logarithmic in the number of hypotheses. We numerically evaluate the performance of our algorithms using both synthetic and real-world DNA mutation data, demonstrating that our algorithms outperform previously proposed heuristic policies by large margins.

## 1 Introduction

Consider the problem of learning the *true* hypothesis from among a (potentially large) set of candidate hypotheses $H$. Assume that the learner is given a (potentially large) set of actions $A$, and knows the distribution of the noisy outcome of each action, under each potential hypothesis. The learner incurs a fixed cost each time an action is selected, and seeks to identify the true hypothesis with sufficient confidence, at minimum total cost. Finally, and most importantly, the learner is allowed to select actions *adaptively*.

This well-studied problem is referred to as *active sequential hypothesis testing*, and as we will describe momentarily, there exists a broad set of results that tightly characterizes the optimal achievable cost under various notions of adaptivity. Unfortunately, the corresponding optimal policies are typically only characterized as the optimal policy to a Markov decision process (MDP)—thus, they remain computationally hard to compute when one requires a policy in practice. This deficiency becomes particularly apparent in modern applications where both the set of hypotheses and set of actions are large. As a concrete example, we will describe later on an application to cancer blood testing that has tens of hypotheses and *billions* of tests at full scale. Thus motivated, *we provide the first approximation algorithms for ASHT*.

We study ASHT under two types of adaptivity: *partial* and *full*, where partial adaptivity requires the sequence of actions to be decided upfront (with adaptively chosen stopping time), and full adaptivity allows the choice of action to depend on previous outcomes. For both problems, we propose *greedy* algorithms that run in $O(|A||H|)$ time, and prove that their expected costs are upper bounded by a non-trivial multiplicative factor of the corresponding optimal costs. Most notably, these

---

\*Equal contribution.

35th Conference on Neural Information Processing Systems (NeurIPS 2021).

| | Noise | Approximation Ratio | Objective | Adaptivity Type |
|---|---|---|---|---|
| [34] | Yes | No | Average | Both |
| [36] | Yes | No | Worst-case | Fully adaptive |
| [25] | No | Yes | Both | Partially adaptive |
| [28, 10] | No | Yes | Both | Fully adaptive |
| [26] | Semi* | No | Both | Both |
| This Work | Yes | Yes | Both | Both |

Table 1: Summary of related work. *Semi refers to a restrictive special case.

approximation guarantees are *independent* of $|A|$ (contrast this with the trivially-achievable guarantee of $O(|A|)$) and *logarithmic* in $|H|$ (the optimal cost itself is often $\Omega(|H|)$).

Our results rely on drawing connections to two existing problems: *submodular function ranking* (SFR) [5] and the *optimal decision tree* (ODT) problem [30]. These connections allow us to tackle what is arguably the primary challenge in achieving approximation results for ASHT, which is its inherent *combinatorial* nature. We will argue that existing heuristics from statistical learning fail precisely because they disregard this combinatorial difficulty—indeed, they largely amount to solving the completely *non-adaptive* version of the problem. At the same time, existing results for SFR and ODT fail to account for *noise* in a manner that would map directly to ASHT—this extension is among our contributions.

**Related Work**   Our work is closely related to three streams of research. Table 1 highlights the key differences between our contributions and those of the most relevant previous works.

(a) **Hypothesis Testing and Asymptotic Performance:** In the classical binary sequential hypothesis testing problem, a decision maker is provided with one action whose outcome is stochastic [42, 3, 31], and the goal is to use the minimum expected number of samples to identify the true hypothesis subject to some given error probability. The ASHT problem, first studied in [14], generalizes this problem to multiple actions. Most related to our work is [34], who formulated a similar problem as an MDP. We will postpone describing and contrasting their work until the experiments section.

(b) **Active Learning and Sample Complexity:** In active learning, the learner is given access to a pool of unlabeled samples (cheaply obtainable) and is allowed to request the label of any sample (expensive) from that pool. The goal is to learn an accurate classifier while requesting as few labels as possible. Some nice surveys include [23] and [39]. Our model extends the classical discrete active learning model [17] in which outcomes are noiseless (deterministic) for any pair of hypothesis and unlabeled sample. When outcomes are noisy, the majority of provable guarantees are provided via sample complexity. [9] showed tight minimax classification error rates for a broad class of distributions. Other sample complexity results on noisy active learning include [43, 36, 7, 4, 24].

(C) **Approximation Algorithms for Decision Trees:** Nearly all optimal approximation algorithms for minimizing cover time are known in the noiseless setting [28, 1, 2]. When the outcome is stochastic, [21] proposed a framework for analyzing algorithms under the *adaptive submodularity* assumption. However, their assumption does not hold for many natural setups including ASHT. [13] considered a variant using ideas from the submodular max-coverage problem, and provided a constant factor approximation to the problem. Other works based on submodular function covering include [35, 22, 29]. [26] provided approximation ratios under the constraint that the algorithm may only terminate when it is completely confident about the outcome.

## 2   Model

We begin by formally introducing the problem. Let $H$ be a finite set of *hypotheses*, among which exactly one is the (unknown) *true* hypothesis that we seek to identify. In this paper, we study the *Bayesian* setting, wherein this true hypothesis is drawn from a known prior distribution $\pi$ over $H$.

Let $A$ be the set of available *actions*. Selecting an action yields a random *outcome* drawn independently from a distribution within a given family $\mathcal{D} = \{D_\theta\}_{\theta \in \Theta}$ of distributions parameterized by

$\Theta \subseteq \mathbb{R}$. We are given a function $\mu : H \times A \to \Theta$ such that if $h \in H$ is the underlying hypothesis and we select action $a \in A$, then the random outcome is drawn independently from distribution $D_{\mu(h,a)}$.[2]

An *instance* of the active sequential hypothesis testing problem is then fully specified by a tuple: $(H, A, \pi, \mu, \mathcal{D})$. The goal is to sequentially select actions to identify the true hypothesis with "sufficiently high" confidence, at minimal expected cost, where cost is measured as the number of actions, and the expectation is with respect to the Bayesian prior and the random outcomes. The notion of *sufficiently high* confidence is encoded by a parameter $\delta \in (0, 1)$, and requires that under any true $h \in H$, the probability of erroneously identifying a different hypothesis is at most $\delta$. An algorithm which satisfies this is said to have achieved $\delta$-**PAC-error**.

We focus on two important families of $D_\theta$'s: the Bernoulli distribution $\text{Ber}(\theta)$ and the Gaussian distribution $N(\theta, \sigma^2)$ where $\sigma^2$ is a known constant (with respect to $\theta$). By re-scaling, without loss of generality we may assume $\sigma^2 = 1$. We require two additional assumptions to state our guarantees. The first assumption is needed for relating the sub-gaussian norm to the KL-divergence, in the partially adaptive version. It ensures that the parameterization $\Theta$ is a meaningful one, in the sense that if $\theta, \theta' \in \Theta$ are far apart, then the distributions $D_\theta$ and $D_{\theta'}$ are also "far" apart (as measured by KL divergence). Assumption 1 is satisfied for $\text{Ber}(\theta)$ when $\theta \in [\theta_{\min}, \theta_{\max}]$ for some constants $0 < \theta_{\min} < \theta_{\max} < 1$, and for $N(\theta, 1)$ where $\theta$ lies in some bounded set in $\mathbb{R}$.

**Assumption 1.** There exist $C_1, C_2 > 0$ such that for any $\theta, \theta' \in \Theta$, we have $C_1 \cdot \text{KL}(D_\theta, D_{\theta'}) \leq (\theta - \theta')^2 \leq C_2 \cdot \text{KL}(D_\theta, D_{\theta'})$, where $\text{KL}(\cdot, \cdot)$ is the Kullback-Leibler divergence.

Our second major assumption simply ensures the existence of a valid algorithm, by assuming that every hypothesis is distinguishable via some action.

**Assumption 2** (Validity). For all $g, h \in H$ where $g \neq h$, there exists $a \in A$ with $\mu(g, a) \neq \mu(h, a)$.

In particular, we do not preclude the possibility that for a given action $a$, there exist (potentially many) pairs of hypotheses $g, h$ such that $\mu(g, a) = \mu(h, a)$. In fact, eliminating such possibilities would effectively wash out any meaningful combinatorial dimension to this problem. On the other hand, any approximation guarantee should be parameterized by some notion of separation (when it exists). For any two hypotheses $g, h \in H$ and any action $a \in A$, define $d(g, h; a) = \text{KL}\left(D_{\mu(g,a)}, D_{\mu(h,a)}\right)$.

**Definition 1** (*s*-separated instance). An ASHT instance is said to be *s*-**separated**, if for any $a \in A$ and $g, h \in H$, $d(g, h; a)$ is either 0 or at least $s$.

Note that in real-world applications, the parameter $s$ could be arbitrarily small, and we introduce the notion of s-separability for the sake of proofs. We will show in Section 6 how our algorithms can easily be modified to handle small $s$ values. In this work, we will study two classes of algorithms that differ in the extent to which adaptivity is allowed.

**Definition 2.** A **fully adaptive** algorithm is a decision tree,[3] each of whose interior nodes is labeled with some action, and each of whose edges corresponds to an outcome. Each leaf is labeled with a hypothesis, corresponding to the output when the algorithm terminates.

**Definition 3.** A **partially adaptive** algorithm $(\sigma, T)$ is specified by a fixed sequence of actions $\sigma = (\sigma_1, \sigma_2, ...)$, with each $\sigma_i \in A$, and a *stopping* time $T$. In particular, under any true hypothesis $h^* \in H$ and for any $t \geq 1$, the event $\{T = t\}$ is independent of the outcomes of actions $\sigma_{t+1}, \sigma_{t+2}, \ldots$ (At the stopping time, the choice of which hypothesis to identify is trivial in our Bayesian setting—it is simply the one with the highest "posterior" probability).

Note that a partially adaptive algorithm can be viewed as a special type of fully adaptive algorithm: it is a decision tree with the additional restriction that the actions at each depth are the same. Therefore, a fully adaptive algorithm may be far cheaper than any partially adaptive algorithm. However, there are many scenarios (e.g., content recommendation and web search [6]) where it is desirable to fix the sequence of actions in advance. Furthermore, in many problems the theoretical analysis of partially adaptive algorithms turns out to be challenging (e.g., [27, 12]).

Thus, given an ASHT instance, there are two problems that we will consider, depending on whether the algorithms are partially or fully adaptive. In both cases, our goal is to design fast approximation

---

[2]In this noisy setting, an action can (and often should) be played for multiple times.

[3]By approximating $D_\theta$'s with discrete distributions, we may assume each node has a finite number of children.

algorithms—ones that are computable in polynomial[4] time and that are guaranteed to incur expected costs at most within a multiplicative factor of the optimum. In the coming sections, we will describe our algorithms and approximation guarantees. Before moving on to this, it is worth noting that our problem setup is extremely generic and captures a number of well-known problems related to decision-making for learning including best-arm identification for multi-armed bandits [8, 19, 32], group testing [18], and causal inference [20], just to name a few.

## 3   Our Approximation Guarantees

We are now prepared to state our approximation guarantees (the corresponding greedy algorithms will be defined in the next two sections). Let $\mathrm{OPT}_\delta^{\mathrm{PA}}$ (resp. $\mathrm{OPT}_\delta^{\mathrm{FA}}$) denote the minimal expected cost of any partially adaptive (resp. fully adaptive) algorithm that achieves $\delta$-PAC-error.

**Theorem 1.** Given an $s$-separated instance and any $\delta \in (0, 1/2)$, there exists a polynomial-time partially adaptive algorithm that achieves $\delta$-PAC-error with expected cost $O\left(s^{-1}\left(1 + \log_{1/\delta}|H|\right)\log\left(s^{-1}|H|\log\delta^{-1}\right)\right)\mathrm{OPT}_\delta^{\mathrm{PA}}$.

To help parse this result, if $\delta$ is on the order of $|H|^{-c}$ for some constant $c$, then the approximation factor becomes $s^{-1}(\log s^{-1} + \log|H| + c\log\log|H|)$.

**Theorem 2.** Given an $s$-separated instance and any $\delta \in (0, 1/2)$, there exists a polynomial-time fully adaptive algorithm that achieves $\delta$-PAC-error with expected cost $O\left(s^{-1}\log\left(|H|\delta^{-1}\right)\log|H|\right)\mathrm{OPT}_\delta^{\mathrm{FA}}$.

A few observations might clarify the significance of these approximation guarantees:

1.  Dependence on action space: Both guarantees are independent of the number of actions $|A|$. This is extremely important since, as described in the Introduction, there exist many applications where the the action space is massive. Moreover, since an approximation factor of $O(|A|)$ is always trivially achievable (by cycling through the actions), instances where $|A|$ is large are arguably the most interesting problems.

2.  Dependence on $|H|, \delta$ and $s$: For fixed $s$ and $\delta$, these are the first polylog-approximations for both partially and fully adaptive versions. Further, for the partially adaptive version, the dependence of the approximation factor on $\delta$ is $O(\log\log\delta^{-1})$ when $\delta^{-1}$ is polynomial in $|H|$, improving upon the naive dependence $O(\log\delta^{-1})$. This is crucial since $\delta$ is often needed to be tiny in practice.

3.  Greedy runtime: While we have only stated in our formal results that our approximation algorithms can be computed in $\mathrm{poly}(|A|, |H|)$ time, the actual time is more attractive: $O(|A||H|)$ for selecting each action. In contrast, the heuristic that we will compare against in the experiments requires solving multiple $\Omega(|A||H|^2)$-sized linear programs.

Despite their similar appearances, Theorems 1 and 2 rely on fundamentally different algorithmic techniques and thus require different analyses. In Section 4, we propose an algorithm inspired by the *submodular function ranking* problem, which greedily chooses a sequence of actions according to a carefully chosen "greedy score." We then sketch the proof of Theorem 1. In Section 5, we introduce our fully adaptive algorithm and sketch the proof of Theorem 2.

Finally, by proving a structural lemma (in Appendix D), we extend the above results to a special case of the **total-error** version (i.e., averaging the error over the prior $\pi$) where the prior is uniform. With $\delta$-*total-error* formally defined in Appendix D:

**Theorem 3.** Given an $s$-separated instance with uniform prior $\pi$ and any $\delta \in (0, 1/4)$, for both the partially and fully adaptive versions, there exist polynomial-time $\delta$-total-error algorithms with expected cost $O\left(s^{-1}\left(1 + |H|\delta^2\right)\log\left(|H|\delta^{-1}\right)\log|H|\right)$ times the optimum.

## 4   Partially Adaptive Algorithm

This section describes our algorithm and guarantee for the partially adaptive problem. We first review necessary background from a related problem, and then state our algorithm (Algorithm 1). Finally, we

---

[4]Throughout this paper, *polynomial time* refers to polynomial in $\left(|H|, |A|, s^{-1}, \delta^{-1}\right)$

sketch the proof of the following more general version of Theorem 1 (complete proof in Appendix B):

**Proposition 1.** Let $\delta \in (0, \frac{1}{4}]$ and consider finding the optimal $\delta$-PAC error algorithm. Given any boosting intensity $\alpha \geq 1$ and coverage saturation threshold $B \in (0, \frac{1}{2} \log \delta^{-1}]$, $\mathrm{RnB}(B, \alpha)$ (as defined in Algorithm 1) produces a partially adaptive algorithm with error $|H| \exp(-\Omega(\alpha B))$ and expected cost $O\left(\frac{\alpha}{s} \log \frac{|H|B}{s}\right) \mathrm{OPT}_\delta^{\mathrm{PA}}$.

By setting $\alpha = 1 + \log_{1/\delta} |H|$ and $B = \frac{1}{2} \log \frac{1}{\delta}$, we immediately obtain Theorem 1.

**Background: Submodular Function Ranking**    In the SFR problem, we are given a ground set $U$ of $N$ *elements*, a family $\mathcal{F}$ of non-decreasing submodular functions $f : 2^U \to [0, 1]$ with $f(U)$ equaling 1 for every $f \in \mathcal{F}$, and a weight function $w : \mathcal{F} \to \mathbb{R}^+$. For any permutation $\sigma = (u_1, ..., u_N)$ of $U$, the *cover time* of $f$ is defined as $\mathrm{CT}(f, \sigma) = \min\{t : f(\{u_1, ..., u_t\}) = 1\}$. The goal is to find a permutation $\sigma$ of $U$ with minimal *cover time* $\sum_{f \in \mathcal{F}} w(f) \cdot \mathrm{CT}(f, \sigma)$. We will use the following greedy algorithm, called GRE, in [5] as a subroutine. The sequence is initialized to be empty and is constructed iteratively. At each iteration, let $S$ be the elements selected so far. GRE selects the element $u$ with the maximal *coverage*, defined as $\mathrm{Cov}(u; S) := \sum_{f \in \mathcal{F}: f(S) < 1} w(f) \cdot (f(S \cup \{u\}) - f(S)) / (1 - f(S))$.

**Theorem 4** ([25]).  For any SFR instance, GRE returns a sequence whose cost is $O(\log \frac{1}{\varepsilon})$ times the optimum, where $\varepsilon := \min\left\{f(S \cup \{u\}) - f(S) > 0 : S \in 2^U, u \in U, f \in \mathcal{F}\right\}$.

**Challenge.**    To motivate our algorithm, consider first the following simple idea: "boost" (or repeat) each action enough, and hence reduce the problem to a deterministic problem $P_{det}$. We then show that the existing technique (submodular function ranking for partially adaptive and greedy analysis for ODT for fully-adaptive) returns a policy with cost $O(\log |H|)$ times the no-noise optimum, and finally show that this no-noise policy can be converted to a noisy version by losing anther factor of $O(\frac{1}{s} \log \frac{|H|}{\delta})$. This analysis was in fact our first attempt. However, there are at least two issues that one runs into:

1. This analysis only compares the policy's cost with the no-noise optimum, but our focus is the $\delta$-noise optimum. In particular, the simpler analysis implicitly assumes that the $\delta$-noise optimum is at least $\Omega(\frac{1}{s} \log \frac{|H|}{\delta})$ times the no-noise optimum, which is not necessarily true. Moreover, it is challenging to analyze the gap between the no-noise optimum and the $\delta$-noise optimum.

2. This simple analysis provides a *weaker* guarantee than ours in terms of $\delta$: it yields a factor of $\log \frac{1}{\delta}$, as opposed to the $\log \log \frac{1}{\delta}$ in our analysis. This distinction is nontrivial, particularly in applications where the error is required to be exponentially small in $|H|$.

**Rank and Boost (RnB) Algorithm**    Our RnB algorithm (Algorithm 1) circumvents the issues above by drawing a connection between ASHT and SFR. First, we observe that although an action is allowed to be selected for multiple times, we may assume each action is selected for at most $M = M(\delta, s, |H|) = O(\frac{|H|^2}{s} \log \frac{|H|}{\delta})$ times. In fact,

**Observation 1.** Let $\widetilde{A}$ be the (multi)-set obtained by creating $M$ copies of each $a \in A$. Then there exists a sequence $\sigma$ of $|\widetilde{A}|$ actions, such that $h^*$, the true hypothesis, has the highest posterior with probability $1 - \delta$ after performing all actions in $\sigma$.

Thus, given $\widetilde{A}$, we define $f_h^B : 2^{\widetilde{A}} \to [0, 1]$ for any coverage saturation level $B > 0$ and $h \in H$ as $f_h^B(S) = \frac{1}{|H|-1} \sum_{g \in H \setminus \{h\}} \min\{1, \frac{1}{B} \sum_{a \in S} d(g, h; a)\}$. One can verify that $f_h^B$ is monotone and submodular. Our algorithm computes a nearly optimal sequence of actions using the greedy algorithm for SFR, and creates a number of copies for each of them. Then we assign a *timestamp* to each $h \in H$, and scan them one by one, terminating when the likelihood of one hypothesis is dominantly high.

**Proof Sketch for Proposition 1.**    We sketch a proof and defer the details to Appendix B. The error analysis follows from standard concentration bounds, so we focus on the cost analysis. Suppose

---

**Algorithm 1 Partially Adaptive Algorithm:** $\mathrm{RnB}(B, \alpha)$

---

1: **Parameters**: Coverage saturation level $B > 0$ and boosting intensity $\alpha > 0$.
2: **Input**: ASHT instance $(H, A, \pi, \mu, \mathcal{D})$
3: **Initialize**: $\sigma \leftarrow \emptyset, \tilde{\sigma} \leftarrow \emptyset$          % Store the selected of actions.
4: **For** $t = 1, 2, ..., |\widetilde{A}|$ **do**        % **Rank:** Compute a sequence of actions.
5:     $S \leftarrow \{\sigma(1), ..., \sigma(t-1)\}$.          % Actions selected so far.
6:     **For** $a \in \widetilde{A}$,          % Compute scores for each action.

$$\mathrm{Score}(a; S) \leftarrow \sum_{h: f_h^B(S) < 1} \pi(h) \frac{f_h^B(S \cup \{a\}) - f_h^B(S)}{1 - f_h^B(S)}.$$

7:     $\sigma(t) \leftarrow \arg\max\{\mathrm{Score}(a; S) : a \in \widetilde{A} \backslash S\}$.     % Select the greediest action.
8: **For** $t = 1, 2, ..., |\tilde{A}|$:        % **Boost:** Repeat each action in $\sigma$ for $\alpha$ times.
9:     **For** $i = 1, 2, ..., \alpha$:
10:     $\tilde{\sigma}\big(\alpha(t-1) + i\big) \leftarrow \sigma(t)$.
11: **For** $t = 1, ..., \alpha|\tilde{A}|$:
12:     Select action $\tilde{\sigma}(t)$ and observe outcome $y_t$.
13:     **If** $t = \alpha \cdot \mathrm{CT}(f_h^B, \sigma)$ for some $h \in H$:     % If $t$ is the *timestamp* for some $h$.
14:        **For** $g \in H \backslash \{h\}$:
15:        $\Lambda(h, g) \leftarrow \prod_{i=1}^{t} \mathbb{P}_{h, \tilde{\sigma}(i)}(y_i) / \mathbb{P}_{g, \tilde{\sigma}(i)}(y_i)$.     % Compute the likelihood ratio.
16:     **If** $\log \Lambda(h, g) \geq \alpha B / 2$ for all $g \in H \backslash \{h\}$, **then** Return $h$.     % Hypothesis identified.

---

$\alpha > 0$, $\delta \in (0, \frac{1}{4})$, and $B \in (0, \frac{1}{2} \log \frac{1}{\delta})$. Let $(\sigma^*, T^*)$ be any optimal partially adaptive algorithm, and let $(\sigma, T)$ be the policy returned by RnB. Our analysis consists of the following steps:

(A) The sequence $\sigma$ does well in covering the submodular functions, in terms of the total cover time: $\sum_{h \in H} \pi(h) \cdot \mathrm{CT}(f_h^B, \sigma) \leq O\left(\log \frac{|H|B}{s}\right) \sum_{h \in H} \pi(h) \cdot \mathrm{CT}(f_h^B, \sigma^*)$.

(B) The expected stopping time of our algorithm is not too much higher than the cover time of its submodular function: $\mathbb{E}_h[T] \leq \alpha \cdot \mathrm{CT}(f_h^B, \sigma)$, $\forall h \in H$.

(C) The expected stopping time in $(\sigma^*, T^*)$ can be lower bounded in terms of the total cover time: $\mathbb{E}_h[T^*] \geq \Omega(s) \cdot \mathrm{CT}(f_h^B, \sigma^*)$, $\forall h \in H$.

Proposition 1 follows by combining the above three steps. In fact,

$$\sum_{h \in H} \pi(h) \cdot \mathbb{E}_h[T] \leq \alpha \sum_{h \in H} \pi(h) \cdot \mathrm{CT}(f_h^B, \sigma) \leq O\left(\alpha \log \frac{|H|B}{s}\right) \sum_{h \in H} \pi(h) \cdot \mathrm{CT}(f_h^B, \sigma^*)$$

$$\leq O\left(\frac{\alpha}{s} \log \frac{|H|B}{s}\right) \sum_{h \in H} \pi(h) \cdot \mathbb{E}_h[T^*],$$

where $\sum_h \pi(h) \cdot \mathbb{E}_h[T]$ is the expected cost of our algorithm, and $\sum_h \pi(h) \cdot \mathbb{E}_h[T^*]$ is the expected cost of the optimal partially adaptive algorithm, i.e. $\mathrm{OPT}_\delta^{\mathrm{PA}}$.

At a high level, (A) can be showed by applying Theorem 4 and observing that the marginal positive increment of each $f_h^B$ is $\Omega(\frac{s}{|H|B})$. Step (B) is obtained from the error analysis. In the key step (C) we fix an arbitrary $\delta$-PAC-error partially adaptive algorithm $(\sigma, T)$ and $h \in H$. Denote $\mathrm{CT}_h = CT(f_h^B, \sigma)$, with $B$ chosen to be $\frac{1}{2} \log \delta^{-1}$. Our goal is to lower bound $\mathbb{E}_h[T]$ in terms of $\mathrm{CT}_h$. To this aim, consider a linear program (LP). Given any $d_1, ..., d_n$, denote $d^i = \sum_{j=1}^i d_j$. Consider

$$LP(d, t): \quad \left\{ \min_z \sum_{i=1}^N i \cdot z_i \ \middle| \ \sum_{i=1}^N d^i z_i \geq \sum_{i=1}^{\mathrm{CT}_h - 1} d_i, \sum_{i=1}^N z_i = 1, z \geq 0. \right\}$$

A feasible solution $z$ can be viewed as a distribution of the stopping time. When $d_i = d(g, h; a_i)$, the first constraint says that the total KL-divergence "collected" at the stopping time has to reach a certain threshold. We show that $z_i = \mathbb{P}_h[T = i]$ is feasible, and the objective value of $z$ is exactly $\mathbb{E}_h[T]$. Hence $\mathbb{E}_h[T]$ is upper bounded by the LP-optimum $LP^*(d, t)$. Finally, we lower bound $LP^*(d, \mathrm{CT}_h - 1)$ by $\Omega(s \cdot \mathrm{CT}_h)$, and the proof follows.

## 5 Fully Adaptive Algorithm

For ease of presentation we only consider the uniform prior version here (though our guarantees do hold for general priors). Our analysis is based on a reduction to the classical ODT problem.

**Background: Optimal Decision Trees** In the ODT problem, an *unknown* true hypothesis $h^*$ is drawn from a set of hypotheses $H$ with some known probability distribution $\pi$. There is a set of known *tests*, each being a (deterministic) mapping from $H$ to a finite *outcome space* set $O$. Thus, when performing a test, we can *rule out* the hypotheses that are inconsistent with the observed outcome, hence reducing the number of *alive* hypotheses. Moreover, the cost $c(T)$ of each test $T$ is known, and the *cost of a decision tree* is defined to be the expected total cost of the tests selected until one hypothesis remains *alive*, in which case we say the true hypothesis is *identified*. The goal is to find a valid decision tree with minimal expected cost.

Note that the ODT problem can be viewed as a special case of the fully adaptive version of our problem where there is no noise and $\delta$ is 0. Consider the following greedy algorithm GRE: let $A$ be the alive hypotheses. Define $\text{Score}(T)$ for each test $T$ to be the minimal (over all possible outcomes) number of alive hypotheses that it rules out in $A$. Then, we select the test $T$ with the highest "bang-per-buck" $\text{Score}(T)/c(T)$. This algorithm is known to be an $O(\log |H|)$-approximation.

**Theorem 5** ([10])**.** For any ODT instance with uniform prior, GRE returns a decision tree whose cost is $O(\log |H|)$ times the optimum.

**Our Algorithm.** We will analyze our greedy algorithm by relating to the above result. Consider the following ODT instance $\mathcal{I}_{\text{ODT}}$ for any given ASHT instance $\mathcal{I}$. The hypotheses set and prior in $\mathcal{I}_{\text{ODT}}$ are the same as in $\mathcal{I}$. For each action $a \in A$, let $\Omega_a := \{\mu(h, a) | h \in H\}$ be the mean outcomes. By Chernoff bound, we can show that when $h$ is the true hypothesis, with high probability the mean outcome is "close" to $\mu(h, a)$ when $a$ is repeated for $c(a)$ times. This motivates us to define a test $T_a : H \to \Omega_a$ s.t. $T_a(h) = \mu(h, a)$, with cost $c(a) = \lceil s(a)^{-1} \log(|H|/\delta) \rceil$, where $s(a) = \min\{d(g, h; a) > 0 : g, h \in H\}$ is the separation parameter under action $a$. Such a test corresponds to selecting $a$ for $c(a)$ times in a row.

For each $\omega \in \Omega_a$, abusing the notation a bit, let $T_a^\omega \subseteq H$ denote the set of hypotheses whose outcome is $\omega$ when performing $T_a$, i.e., $T_a^\omega = \{h : \mu(h, a) = \omega\}$. At each step, Algorithm 2 selects an action $\hat{a}$ using the greedy rule (Step 4) and then repeat $\hat{a}$ for $c(\hat{a})$ times. Then we round the empirical mean of the observations to the closest element $\hat{\omega}$ in $\Omega_a$, ruling out inconsistent hypotheses, i.e., the $h$'s with $\mu(h, a) \neq \hat{\omega}$. We terminate when only one hypothesis remains alive.

**Analysis** We sketch a proof for Theorem 2 and defer the details to Appendix C. Let $h^*$ be the true hypothesis. By Hoeffding's inequality, in each iteration, with probability $1 - e^{-\log(|H|/\delta)} = 1 - \frac{\delta}{|H|}$ it holds $\hat{\omega} = \mu(h^*, \hat{a})$. Since in each iteration, $|H|$ decreases by at least 1, there are at most $|H| - 1$ iterations. Thus by union bound, the total error is at most $\delta$.

Next we analyze the cost. Let GRE be the cost of Algorithm 2 and $\text{ODT}^*$ be the optimum of the ODT instance $\mathcal{I}_{\text{ODT}}$. For the sake of analysis, we consider a "fake" cost $c' := \lceil s^{-1} \log(|H|/\delta) \rceil$, which does not depend on $a$. The definition of the ODT instance $\mathcal{I}_{\text{ODT}}$ remains the same except that each test has *uniform* cost $c'$ (as opposed to $c(a)$). Let $c(T)$ and $c'(T)$ be the costs of the greedy tree $T$ returned by Algorithm 2 under $c$ and $c'$ respectively. Then by Theorem 5, $c'(T) \leq O(\log |H|) \cdot \text{ODT}^*$. Note that $c' \leq c(a)$ for each $a$ since the separation parameter $s$ is no larger than $s(a)$ by definition. Hence,

$$\text{GRE} = c(T) \leq c'(T) \leq O(\log |H|) \cdot \text{ODT}^*. \tag{1}$$

We relate $\text{ODT}^*$ to $\text{OPT}_\delta^{FA}$ using the following result (see proof in Appendix C):

**Proposition 2.** $\text{ODT}^* \leq O(\frac{1}{s} \log \frac{|H|}{\delta}) \cdot \text{OPT}_\delta^{FA}$.

The above is established by showing how to convert a $\delta$-PAC-error fully adaptive algorithm to a valid decision tree, using only tests in $\{T_a\}$, and inflating the cost by a factor of $O(\frac{1}{s} \log \frac{|H|}{\delta})$. Combining Proposition 2 with Equation (1), we obtain $GRE \leq O(\frac{1}{s} \log \frac{|H|}{\delta} \log |H|) \cdot \text{OPT}_\delta^{FA}$.

Finally we remark that this analysis can easily be extended to general priors by reduction to the *adaptive submodular ranking* (ASR) problem [35], which captures ODT as a special case. One may

---
**Algorithm 2 Fully Adaptive Algorithm**
---
1: **Input:** ASHT instance $(H, A, \pi, \mu, \mathcal{D})$ and error $\delta \in (0, 1/2)$.
2: $H_{\text{alive}} \leftarrow H$.      % *Alive* hypotheses.
3: **while** $|H_{\text{alive}}| \geq 2$ **do**
4:      $\widehat{a} \leftarrow \arg\max_{a \in A} \left\{ \min_{\omega \in \Omega_a} |H_{\text{alive}} \backslash T_a^\omega| \right\}$.      % Greedy step.
5:      $c(\widehat{a}) \leftarrow \lceil s(\widehat{a})^{-1} \log(|H|/\delta) \rceil$.      % # times to boost for sufficient confidence.
6:      Select $\widehat{a}$ for $c(\widehat{a})$ times consecutively and observe outcomes $X_1, ..., X_{c(\widehat{a})}$.
7:      $\widehat{\mu} \leftarrow \sum_{i=1}^{c(\widehat{a})} X_i$.      % Mean outcome.
8:      $\widehat{\omega} \leftarrow \arg\min\{|\widehat{\mu} - \omega| : \omega \in \Omega_a\}$.      % Round $\widehat{\mu}$ to the closest $\omega$.
9:      $H_{\text{alive}} \leftarrow H_{\text{alive}} \cap T_{\widehat{a}}^{\widehat{\omega}}$.      % Update the alive hypotheses.
10: **end while**
---

easily verify that the main theorem in [35] implies that a (slightly different) greedy algorithm achieves $O(\log |H|)$-approximation for the ODT problem with general prior, test costs, and an arbitrary number of branches in each test. Thus for general prior, the same analysis goes through if we first reduce ASHT to ASR, and then replace the greedy step (Step 4 in Algorithm 2) with the greedy criterion for ASR.

## 6 Experiments

Although our theoretic guarantees depend on the separability parameters $s$ (which was introduced by the boosting steps), in this section, we numerically demonstrate that with small modifications our algorithms perform well when $s$ is small on both synthetic and real-world data. Our primary benchmarks are a polynomial-time policy proposed by [34] (*Policy 1*[5]) and a completely random policy. To our knowledge, the policy proposed by [34] is the state-of-art algorithm (with theoretical guarantees) that can be applied to our problem setup. The rest of this section is organized as follows: first, we describe the benchmark policies and the implementation of our own policies. Then in Section 6.1, we describe the setup and results of our synthetic experiments. Finally, in Section 6.2, we test the performance of our fully adaptive algorithm on a publicly-available dataset of genetic mutations for cancer—COSMIC [40, 16].

**Algorithm Details** In all algorithms, we start with a uniform prior, and update our prior distribution (over the hypotheses space) each time an observation is revealed. Unless otherwise mentioned, the algorithm terminates if the posterior probability of a hypothesis is above the threshold $1 - \delta$.

**Random Baseline** At each step, an action was uniformly chosen from the set of all actions.

**Partially Adaptive** We implement the partially adaptive algorithm described in Section 4, with the modifications that 1) the amount of boosting is now a built-in feature of the algorithm, and 2) breaking ties according to some heuristic. We describe the modified algorithm in Appendix E.

**Fully Adaptive** We implement our algorithm described in Section 5, with the modifications that 1) the amount of boosting is considered as a tunable parameter, 2) a hypothesis is only considered to be ruled out when we are deciding which action to perform, 3) we do not boost if no action can further distinguish any hypotheses in the alive set, 4) we break ties according to some heuristic. In particular, Modification 1) addresses the issues that our fully adaptive algorithm in Section 5 over-boosts. Modification b) controls the error probability $\delta$ when we decrease the amount of boosting. Modification c) handles small $s$ without increasing the boosting factor. We formally describe this modified algorithm in Appendix F.

**NJ Algorithm.** *NJ Adaptive* [34] is a two-phase algorithm that solves a relaxed version of our problem, where the objective is to minimize a weighted sum of the expected number of tests and the likelihood of identifying the wrong hypothesis, i.e., $\min \mathbb{E}(T) + Le$, where $T$ is the termination time, $L$ is the penalty for a wrong declaration, and $e$ is the probability of making that wrong declaration. The problem was formulated as a Markov decision process whose state space is the posterior distribution over the hypotheses. In Phase 1, which lasts as long as the posterior probability of all hypotheses is below a carefully chosen threshold, the action is sampled according to a distribution that is selected

---

[5]*Policy 2* in [34] does not have asymptotic guarantees and so is not considered in our experiments.

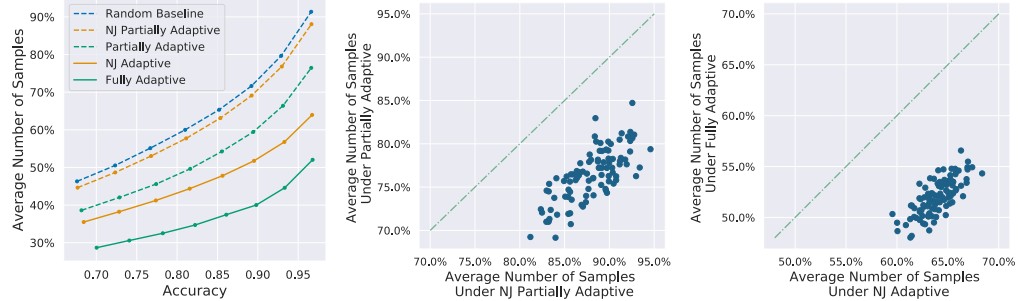

Figure 1: Comparison of our fully and partially adaptive algorithms with *NJ Adaptive*, *NJ Partially Adaptive* and *Random Baseline* on synthetic data. The average number of samples is normalized with respect to the largest number of sample required in *Random Baseline*. Left: each dot corresponds to the average performance of 100 randomly generated instances each averaged over 2,000 replications. Middle and Right: contains the same 100 instances in the left figure. Each dot corresponds to one instance and each averaged over 2,000 replications. Middle and Right: the average accuracies of those 100 instances in all algorithms equal to 0.97.

to maximize the minimum expected KL divergence among all pairs of outcome variables. In Phase 2, when one of the hypotheses has posterior probability above the chosen threshold, $r$, the action is sampled according to a distribution selected to maximize the minimum expected KL divergence between the outcome of this hypothesis and the outcomes of all other hypotheses. This threshold was optimized over in both synthetic and real-world experiments. The algorithm stops if the posterior of a hypothesis is above the threshold $1 - L^{-1}$. *NJ Partially Adaptive* contains only the Phase 1 policy.

## 6.1 Synthetic Experiments

**Parameter Generation and Setup**     Figure 1 summarizes the results of our partially and fully adaptive experiments on synthetic data. Both figures were generated with 100 instances: each with 25 hypotheses and 40 actions. The outcome of each action under each hypothesis is binary, i.e., the $D_{\mu(h,a)}$'s are the Bernoulli distributions, where $\mu(a, h)$ were *uniformly* sampled from the [0,1] interval. Each instance was then averaged over 2,000 replications, where a "ground truth" hypothesis was randomly drawn. The prior distribution, $\pi$, was initialized to be uniform for all runs. On the horizontal axis, the accuracies of both algorithms were averaged over these 100 instances, where the accuracy is calculated as the percentage of correctly identified hypotheses among the 2,000 replications. On the vertical axis, the number of samples used by the algorithm is first averaged over the 2,000 replications and then averaged over the 100 instances.

**Results**     In Figure 1 (left), we observe that 1) the performance of our fully adaptive algorithm dominates those of all other algorithms, 2) our partially adaptive algorithm outperforms all other partially adaptive algorithms, and 3) the performance of adaptive algorithms outperform those of partially adaptive algorithms. The threshold for entering Phase 2 policy in *NJ Adaptive* was set to be 0.1. Indeed, we observe that *NJ Adaptive* outperforms *NJ Partially Adaptive*. In Figure 1 (middle), $\delta$ equals to 0.05 for both *NJ Partially Adaptive* and *Partially Adaptive*. We observe that our partially and fully adaptive algorithms outperform *NJ Partially Adaptive* and *NJ Adaptive* instance-wise by large margins respectively in Figure 1 middle and left.

## 6.2 Real-World Experiments

**Problem Setup**     Our real-world experiment is motivated by the design of DNA-based blood tests to detect cancer. In such a test, genetic mutations serve as potential signals for various cancer types, but DNA sequencing is, even today, expensive enough that the 'amount' of DNA that can be sequenced in a single test is limited if the test is to remain cost-effective. For example, one of the most-recent versions of these tests [15] involved sequencing just 4,500 *addresses* (from among 3 billion total addresses in the human genome), and other tests have had similar scale (e.g., [38, 11, 37]). Thus, one promising approach to the ultimate goal of a cost-effective test is adaptivity.

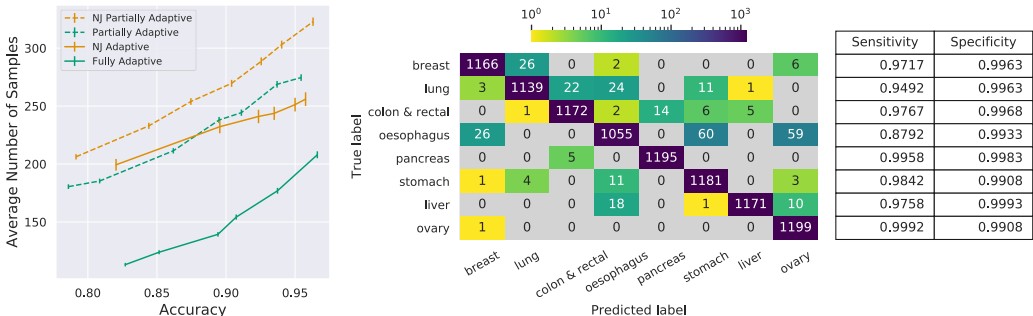

Figure 2: Comparison of our partially and fully adaptive algorithms with those of NJ's on real-world data—COSMIC. Left: each point is averaged over 9,600 replications. The error bars are the 95 percentage confidence intervals for the estimated means. Middle: the confusion matrix of *Fully Adaptive* where the algorithm accuracy equals to 0.97, and each row sums up to 1,200. Left: the sensitivity and specificity our algorithm (middle) for each cancer type.

Our experiments are a close reproduction of the setup used by [15] to identify their 4,500 addresses. We use genetic mutation data from real cancer patients: the publicly-available *catalogue of somatic mutations in cancer* (COSMIC) [40, 16], which includes the de-identified gene-screening panels for 1,350,015 patients. We treated 8 different types of cancer (as indicated in [15]) as the 8 hypotheses, and identified 1,875,408 potentially mutated genetic addresses. To extract the tests, we grouped the the genetic addresses within an interval of 45 (see [15] for the biochemical reasons behind this choice), resulting in 581,754 potential tests. We then removed duplicated tests (i.e., the tests that share the same outcome distribution for all 8 cancer types), resulting in 23,135 final tests that we consider in our experiments. Note that the duplicated tests can be removed here since they are exchangeable in our problem setting. However, the set of final tests might be different for a different set of ground-truth cancer types. From the data, we extracted a "ground-truth" table of mutation probabilities containing the likelihood of a mutation in any of the 23,135 genetic address intervals being found in patients with any of the 8 cancer types. This served as the instance for our experiment. The majority of the mutation probabilities in our instances was either zero or some small positive number. To calculate the KL divergence between these probabilities, we replace zero with the number $10^{-10}$ in our instance.

**Results.** Although in reality, all patients have different priors for having different cancers, in our experiments, we assume that the truth hypothesis (cancer type) was drawn uniformly, and we initialize uniform priors for all algorithms. Similar to Figure 1, in Figure 2 (left) we observe that 1) the performance of our algorithm dominates those to the rest algorithms, and 2) our partially adaptive algorithm outperforms *NJ Partially Adaptive*. However, unlike Figure 1, we observe that *NJ Adaptive* underperforms *Partially Adaptive* when the accuracies are low on this instance. The threshold for entering Phase 2 policy, $r$, in *NJ Adaptive* was set to be 0.3. Since Phase 1 policy is less efficient than Phase 2 policy, we observe that the performance of *NJ Adaptive* is convex with respect to $r$—when $r$ is small, the algorithm is more likely to alternate between Phases 1 and 2 policies and when $r$ is large we spend more time in Phase 1 policy. As a result, we observe that variance of *NJ Adaptive* is relatively high when compared with those of other algorithms. Note that due to the nature of the sparsity of our instance, the performance of the random baseline was very poor when compared with these of *NJ Adaptive* and *Fully Adaptive* and thus was excluded. Figure 2 (middle) is the confusion matrix corresponding to our fully adaptive algorithm where the algorithm accuracy equals to 0.97, and Figure 2 (right) corresponds to the sensitivity and specificity of our algorithm for each cancer type (in the same ordering as) in the middle figure.

## 7   Conclusions

In this work we provided the first approximation guarantees for the ASHT problem and demonstrated the efficiency of the proposed algorithms through numerical experiments on genetic mutation data. Under the current framework, it is challenging to improve the $s^{-1}$ dependence on $s$ in the approximation factors, since we have to boost each action for $s^{-1}$ times to apply the concentration bounds. However, in our numerical experiments, by reducing the number of times for boosting, we achieved better performance when compared with existing heuristics.

## Acknowledgments and Disclosure of Funding

We would like thank the anonymous reviewers for their careful reviews, and we declare no conflict of interests.

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
