# A Prerequisite: Subgaussian Random Variables

We will consider the commonly used subgaussian distributions ([41]). Loosely speaking, a random variable is subgaussian if its tail vanishes at a rate faster than some Gaussian distributions.

**Definition 4** (Subgaussian norm). Let $X$ be a random variable, its *subgaussian norm* is defined as $\|X\|_{\psi_2} := \inf\{t : \mathbb{E}[e^{X^2/t^2}] \leq 2\}$. Moreover, $X$ is called *subgaussian* if $\|X\|_{\psi_2} < \infty$.

Many commonly used distributions satisfy this assumption, e.g., Bernoulli, uniform, and Gaussian distributions. We introduce a standard concentration bound for subgaussian random variables.

**Theorem 6** (Hoeffding Inequality [41]). Let $X_1, ..., X_n$ be independent subgaussian random variables. Then for any $\eta > 0$, it holds that

$$\mathbb{P}\left[\left|\sum_{i=1}^{n} X_i - \sum_{i=1}^{n} \mathbb{E}(X_i)\right| \geq \eta\right] \leq 2\exp\left(-\frac{2\eta^2}{\sum_{i=1}^{n}\|X_i\|_{\psi_2}^2}\right).$$

To show the correctness of our algorithm, we need to consider the *log-likelihood ratio* (LLR), formally defined as follows:

**Definition 5.** For any $a \in A$ and $h, g \in H$, define $Z(h, g; a) = \log \frac{\mathbb{P}_{h,a}(\xi)}{\mathbb{P}_{g,a}(\xi)}$ where $\xi \sim D_{\mu(h,a)}$.

We will assume that the subgaussian norm of the LLR between two hypotheses is not too large when compared to the difference of their parameters, as formalized below:

**Definition 6.** Let $\rho > 0$ be the minimal number s.t. for any pair of distinct hypotheses $h, g \in H$ and action $a \in A$, it holds that $\|Z(h, g; a)\|_{\psi_2} \leq \rho \cdot |\mu(g, a) - \mu(h, a)|$.

We will present an error analysis for general $\rho$. Prior to that, we first point out that many common distributions satisfy $\rho = O(1)$.

**Examples.** It is straightforward to verify that $\rho = O(1)$ for the following common distributions:

- Bernoulli distributions: $D_\theta = Ber(\theta)$ where $\theta \in [\theta_{min}, \theta_{max}]$ for constants $\theta_{min}, \theta_{\max} \in (0, 1)$, and
- Gaussian distributions: $D_\theta = N(\theta, 1)$ where $\theta \in [\theta_{min}, \theta_{max}]$ for constants $\theta_{min} < \theta_{\max}$.

Take Bernoulli distribution as an example. Fix any hypotheses $h, g \in H$ and action $a \in A$, write $\Delta = \mu(h, a) - \mu(g, a)$. Then, $Z = Z(h, g; a)$ can be rewritten as

$$Z = \begin{cases} \log(1 + \frac{\Delta}{\mu(g,a)}), & \text{w.p. } \mu(h, a), \\ \log(1 - \frac{\Delta}{1-\mu(g,a)}), & \text{w.p. } 1 - \mu(h, a). \end{cases}$$

Since $0 < \theta_{min} \leq \mu(g, a) \leq \theta_{\max} < 1$, we have $|Z| \leq C|\Delta|$ almost surely where $C = 2\max\{(1 - \theta_{max})^{-1}, \theta_{min}^{-1}\}$. Moreover, it is known that (see [41]) any subgaussian random variable $Z$ satisfies $\|Z\|_{\psi_2} \leq \frac{1}{\ln 2}\|Z\|_\infty$, so it follows that

$$\|Z\|_{\psi_2} \leq \frac{1}{\ln 2}\|Z\|_\infty \leq \frac{C\Delta}{\ln 2} = O(\Delta).$$

Thus $\rho = O(1)$.

# B Proof of Proposition 1

## B.1 Error Analysis

We first prove that at each timestamp $\tau(h)$, with high probability our algorithm terminates and returns $h$.

**Lemma 1.** Let $B > 0$. If $h \in H$ is the true hypothesis, then w.p. $1 - e^{-\Omega(\rho^{-2}\alpha B)}$, it holds $\log \Lambda(h, g; \tau(h)) \geq \frac{1}{2}\alpha B$ for all $g \neq h$.

*Proof.* Let $\tilde{\sigma} = (a_1, a_2, ...)$ be the sequence *after* the boosting step, so $a_1 = ... = a_\alpha, a_{\alpha+1} = ... = a_{2\alpha}$, so on so forth. Write $Z_i = Z(h, g; a_i)$, then for any $t \geq 1$, it holds $\log \Lambda(h, g; t) = \sum_{i=1}^{t} Z_i$. By the definition of cover time, $\sum_{i=1}^{\tau(h)} d(h, g; a_i) = \sum_{i=1}^{\tau(h)} \mathbb{E}(Z_i) \geq \alpha B$. Thus,

$$\mathbb{P}_h \left[ \log \Lambda(h, g; \tau(h)) < \frac{1}{2}\alpha B \right] = \mathbb{P}_h \left[ \sum_{i=1}^{\tau(h)} Z_i < \frac{1}{2}\alpha B \right]$$

$$\leq \mathbb{P}_h \left[ \left| \sum_{i=1}^{\tau(h)} Z_i - \sum_{i=1}^{\tau(h)} \mathbb{E}(Z_i) \right| > \frac{1}{2} \sum_{i=1}^{\tau(h)} \mathbb{E}(Z_i) \right]. \qquad (2)$$

By Theorem 6,

$$\text{Equation (2)} \leq \exp\left( -\Omega\left( \frac{(\alpha B)^2}{\sum_{i=1}^{\tau(h)} \|Z_i\|_{\psi_2}^2} \right) \right). \qquad (3)$$

We next show that $\sum_{i=1}^{\tau(h)} \|Z_i\|_{\psi_2}^2 \leq O(\rho^2 \alpha B)$. Write $\Delta_i = \mu(h, a_i) - \mu(g, a_i)$, then by Assumption 2, $\Delta_i^2 \leq C_2 \cdot d(h, g; a_i)$. Note that $\|Z_i\|_{\psi_2} \leq \rho \Delta_i$, so it follows that

$$\sum_{i=1}^{\tau(h)} \|Z_i\|_{\psi_2}^2 \leq \rho^2 \sum_{i=1}^{\tau(h)} \Delta_i^2 \leq C_2 \rho^2 \sum_{i=1}^{\tau(h)} d(h, g; a_i). \qquad (4)$$

Recall that $\sigma$ is the sequence *before* boosting. Write $t = CT(f_h^B, \sigma)$ for simplicity. By the definition of cover time,

$$\sum_{i=1}^{\alpha t} d(h, g; a_i) \geq \alpha B \geq \sum_{i=1}^{\alpha(t-1)} d(h, g; a_i).$$

Note that $\tau(h) = \alpha t$, so

$$\sum_{i=1}^{\alpha t} d(h, g; a_i) \leq 2 \sum_{i=1}^{\alpha(t-1)} d(h, g; a_i) \leq 2\alpha B.$$

Combining the above with Equation (4), we have

$$\sum_i \|Z_i\|_{\psi_2}^2 \leq 2C_2 \rho^2 \alpha B.$$

Substituting into Equation (3), we obtain

$$\mathbb{P}_h \left[ \log \Lambda(h, g; \tau(h)) < \frac{1}{2}\alpha B \right] \leq e^{-\Omega(\rho^{-2}\alpha B)}.$$

The proof completes by applying the union bound over all $g \in H \backslash \{h\}$. $\qquad \square$

By a similar approach we may also show that it is unlikely that the algorithm terminates at a wrong time stamp before scanning the correct one.

**Lemma 2.** Let $B > 0$. If $h \in H$ is the true hypothesis, then for any $g \neq h$, it holds that $\log \Lambda(g, h; \tau(g)) < \frac{1}{2}\alpha B$ with probability $1 - e^{-\Omega(\rho^{-2}\alpha B)}$.

We are able to bound the error of the RnB algorithm by combining Lemma 1 and Lemma 2.

**Proposition 3.** For any true hypothesis $h \in H$, algorithm $RnB(B, \alpha)$ returns $h$ with probability at least $1 - |H|e^{-\Omega(\rho^{-2}\alpha B)}$. In particular, if the outcome distribution $D_\mu$ is $Ber(\mu)$, then $\rho = O(1)$ and the above probability becomes $1 - |H|e^{-\Omega(\alpha B)}$.

## B.2 Cost Analysis

Recall that in Section 4, only Step (C) remains to be shown, which we formally state below.

**Proposition 4.** Let $(\sigma, T)$ be a $\delta$-PAC-error partially adaptive algorithm. For any $B \leq \log \delta^{-1}$ and $h \in H$, it holds that $\mathbb{E}_h[T] \geq \Omega\big(s \cdot \mathrm{CT}(f_h^B, \sigma)\big)$.

We fix an arbitrary $h \in H$ and write $\mathrm{CT}_h := \mathrm{CT}(f_h^B, \sigma)$, where we recall that $\sigma$ is the sequence of actions before boosting (do not confuse with $\tilde{\sigma}$). To relate the stopping time $T$ (under $h$) to the cover time of the submodular function for $h$ in $\sigma$, we introduce a linear program. We will show that for suitable choice of $d$, we have

- $LP^*(d, \mathrm{CT}_h - 1) \leq \mathbb{E}_h[T]$, and
- $LP^*(d, \mathrm{CT}_h - 1) \geq \Omega(s \cdot \mathrm{CT}_h)$.

Hence proving Step (C) in the high-level proof sketched in Section 4.

We now specify our choice of $d$. For any $d_1, ..., d_N \in \mathbb{R}_+$, write $d^t := \sum_{i=1}^t d_i$ for any $t$ and consider

$$
LP(d, t): \quad \min_z \sum_{i=1}^N i \cdot z_i
$$
$$
s.t. \sum_{i=1}^N d^i z_i \geq d^t,
$$
$$
\sum_{i=1}^N z_i = 1,
$$
$$
z \geq 0.
$$

We will consider the following choice of $d_i$'s. Suppose $(\sigma, T)$ has $\delta$-PAC-error where $\delta \in (0, 1/4]$. For any pair of hypotheses $h, g$ and any set of actions $S$, define

$$
K_{h,g}^B(S) = \min\left\{1, B^{-1} \sum_{a \in S} d(h, g; a)\right\}.
$$

Hence,

$$
f_h^B(S) = \frac{1}{|H| - 1} \sum_{g \in H \setminus \{h\}} K_{h,g}^B(S).
$$

Fix any $B \leq \log \delta^{-1}$ and let $g$ be the last hypothesis separated from $h$, i.e.,

$$
g := \arg \max_{h' \in H \setminus \{h\}} \left\{\mathrm{CT}(K_{h,h'}^B, \sigma)\right\}.
$$

Then by the definition of cover time, we have $\mathrm{CT}_h = \mathrm{CT}(f_h^B, \sigma) = \mathrm{CT}(K_{hg}^B, \sigma)$. Without loss of generality,[6] we assume that all actions $a$ satisfy $\mu(h, a) = \mu(g, a)$ in $\tilde{\sigma} = (a_1, .., a_N)$. We choose the LP parameters to be $d_i = d(h, g, a_i)$ for $i \in [N]$.

**Outline.** We will first show that the LP optimum is upper bounded by the expected termination time $T$ (Proposition 5). We then lower bound it in terms of $\mathrm{CT}_h$ (Proposition 6).

**Proposition 5.** Suppose $(\sigma, T)$ has $\delta$-PAC-error for some $0 < \delta \leq \frac{1}{4}$. Let $z_i = \mathbb{P}_h[T = i]$ for $i \in [N]$, then $z = (z_1, ..., z_N)$ is feasible to $LP(d, \mathrm{CT}_h - 1)$.

Note that $\mathbb{E}_h(T)$ is simply the objective value of $z$, thus Proposition 5 immediately implies:

**Corollary 1.** $\mathbb{E}_h(T) \geq LP^*(d, \mathrm{CT}_h - 1)$.

We next lower bound the expected log-likelihood when the algorithm stops.

---

[6]If there is some action $a$ with $d(h, g; a) = 0$, then we simply remove it. This will not change the argument.

**Lemma 3.** [36] Let $\mathbb{A}$ be any algorithm (not necessarily partially adaptive) for the ASHT problem. Let $h, g \in H$ be any pair of distinct hypotheses and $O$ be the random output of $\mathbb{A}$. Define the error probabilities $P_{hh} = \mathbb{P}_h(O = h)$ and $P_{hg} = \mathbb{P}_h(O = g)$. Let $\Lambda$ be the likelihood ratio between $h$ and $g$ when $\mathbb{A}$ terminates. Then,

$$\mathbb{E}_h(\log \Lambda) \geq P_{hh} \log \frac{P_{hh}}{P_{hg}} + (1 - P_{hh}) \log \frac{1 - P_{hh}}{1 - P_{hg}}.$$

*Proof.* Let $\mathcal{E}$ be the event that the output is $h$. Then by Jensen's inequality, we have

$$\mathbb{E}_h(\log \Lambda_T | \mathcal{E}) \geq -\log \mathbb{E}_h\left(\Lambda^{-1} | \mathcal{E}\right) = -\log \frac{\mathbb{E}_h\left(\mathbb{1}(\mathcal{E}) \cdot \Lambda^{-1}\right)}{\mathbb{P}_h(\mathcal{E})}. \tag{5}$$

Recall that an algorithm can be viewed as a decision tree in the following way: each internal node is labeled with an action, and each edge below it corresponds to a possible outcome; each leaf corresponds to termination and is labeled with a hypothesis corresponding to the output. Write $\sum_x$ as the summation over all leaves and let $p_h(x)$ (resp. $p_g(x)$) be the probability that the algorithm terminates in leaf $x$ under $h$ (resp. $g$), then,

$$
\begin{aligned}
\mathbb{E}_h\left(\mathbb{1}(\mathcal{E}) \cdot \Lambda^{-1}\right) &= \sum_x \mathbb{1}(x \in \mathcal{E}) \cdot \Lambda^{-1}(x) \cdot p_h(x) \\
&= \sum_x \mathbb{1}(x \in \mathcal{E}) \cdot \frac{p_g(x)}{p_h(x)} p_h(x) \\
&= \sum_x \mathbb{1}(x \in \mathcal{E}) \cdot p_h(x) \\
&= \mathbb{E}_h(\mathbb{1}(x \in \mathcal{E})) = P_{hg}.
\end{aligned}
$$

Combining the above with Equation (5), we obtain

$$\mathbb{E}_h(\log \Lambda | \mathcal{E}) \geq \log \frac{P_{hh}}{P_{hg}}.$$

Similarly, we have $\mathbb{E}_h\left(\log \Lambda | \bar{\mathcal{E}}\right) \geq \log \frac{1 - P_{hh}}{1 - P_{hg}}$, where $\bar{\mathcal{E}}$ is the event that the output is not $h$. The proof follows immediately by combining these two inequalities. $\square$

To show Proposition 5, we need a standard concept—stopping time.

**Definition 7** (Stopping time [33]). Let $\{X_i\}$ be a sequence of random variables and $T$ be an integer-valued random variable. If for any integer $t$, the event $\{T = t\}$ is independent with $X_{t+1}, X_{t+2}, ...$, then $T$ is called a **stopping time** for $X_i$'s.

**Lemma 4** (Wald's Identity). Let $\{X_i\}_{i \in \mathbb{N}}$ be independent random variables with means $\{\mu_i\}_{i \in \mathbb{N}}$, and let $T$ be a stopping time w.r.t. $X_i$'s. Then, $\mathbb{E}\left(\sum_{i=1}^T X_i\right) = \mathbb{E}\left(\sum_{i=1}^T \mu_i\right)$.

**Proof of Proposition 5.** One may verify that the lower bound in Lemma 3 is increasing w.r.t $P_{hh}$ and decreasing w.r.t $P_{hg}$. Therefore, since $\mathbb{A}$ has $\delta$-PAC-error, by Lemma 3 it holds that

$$\mathbb{E}_h\left(\log \Lambda(h, g; T)\right) \geq (1 - \delta) \log \frac{1 - \delta}{\delta} + \delta \log \frac{\delta}{1 - \delta} \geq \frac{1}{2} \log \frac{1}{\delta} \geq B \geq d^{\mathrm{CT}_h - 1}.$$

By Lemma 4,

$$\sum_{i=1}^N d^i z_i = \sum_{i=1}^N d^i \cdot \mathbb{P}_h(T = i) = \mathbb{E}_h\left(\log \Lambda(h, g; T)\right).$$

The proof follows by combining the above. $\square$

So far we have upper bounded $LP^*(d, \mathrm{CT}_h - 1)$ using $\mathbb{E}_h(T)$. To complete the proof, we next lower bound $LP^*(d, \mathrm{CT}_h - 1)$ by $\Omega(s \cdot \mathrm{CT}_h)$.

**Lemma 5.** $LP^* = \min_{i \leq t < j} LP^*_{ij}$ where $LP^*_{ij} = i + (j - i) \frac{d^t - d^i}{d^j - d^i}$.

*Proof.* Observe that for any optimal solution, the inequality constraint must be tight. By linear algebra, we deduce that any basic feasible solution has support size two.

Consider the solutions whose only nonzero entries are $i, j$. Then, $LP(d, t)$ becomes

$$LP_{ij}(d, t) : \quad \min_{z_i, z_j} \quad iz_i + jz_j$$
$$\text{s.t. } d^i z_i + d^j z_j = d^t,$$
$$z_i + z_j = 1,$$
$$z \geq 0.$$

Note that since $d^i < d^j$, $LP_{i,j}(d, t)$ admits exactly one feasible solution, whose objective value can be easily verified to be $LP_{ij}^* := i + (j - i)\frac{d^t - d^i}{d^j - d^i}$. $\qquad \square$

Now we are ready to lower bound the LP optimum.

**Proposition 6.** For any $d = (d_1, ..., d_N) \in \mathbb{R}^N$ and $t \in \mathbb{N}$, it holds that $LP^*(d, t) \geq t \cdot \min\{d_i\}_{i \in [N]}$.

*Proof.* By Lemma 5, it suffices to show that $LP_{ij}^* \geq d^t$ for any $i \leq t < j$. Since $d^k < k$ for any integer $k$,
$$(j - d^t)(d^t - d^i) \geq (d^j - d^t)(d^t - i).$$

Rearranging, the above becomes
$$i(d^j - d^i) + (j - i)(d^t - d^i) \geq d^t(d^j - d^i),$$

i.e.,
$$i + (j - i)\frac{d^t - d^i}{d_j - d^i} \geq d^t.$$

Note that the LHS is exactly $LP_{ij}^*$, thus $LP^*(d, t) \geq d^t \geq t \cdot \min\{d_i\}_{i \in [N]}$ for any $t \in \mathbb{N}$. $\qquad \square$

It immediately follows that $LP^*(d, t) \geq st$, completing the proof of Proposition 1.

## C  Proof of Proposition 2

We first formally define a decision tree, not only for mathematical rigor but more importantly, for the sake of introducing a novel variant of ODT. Recall that $\Omega$ is the space of the test outcomes, which we assume to be discrete for simplicity.

**Definition 8** (Decision Trees). A decision tree is a rooted tree, each of whose interior (i.e., non-leaf) node $v$ is associated with a *state* $(A_v, T_v)$, where $T_v$ is a test and $A_v \subseteq H$. Each interior node has $|\Omega|$ children, each of whose edge to $v$ is labeled with some outcome. Moreover, for any interior node $v$, the set of *alive* hypotheses $A_v$ is the set of hypotheses consistent with the outcomes on the edges of the path from the root to $v$. A node $\ell$ is a *leaf* if $|A_\ell| = 1$. The decision tree terminates and outputs the only alive hypothesis when it reaches a leaf.

To relate $OPT_\delta^{FA}$ to the optimum of a suitable ODT instance, we introduce a novel variant of ODT. As opposed to the ordinary ODT where the output needs to be correct with probability 1, in the following variant, we consider decision trees which may *err* sometimes:

**Definition 9** (Incomplete Decision Tree). An *incomplete decision tree* is a decision tree whose leaves $\ell$'s are associated with *state*s $(A_\ell, p_\ell)$'s, where $A_\ell$ represents the subset of hypotheses consistent with all outcomes so far, and $p_\ell$ is a distribution over $A_\ell$. A hypothesis is randomly drawn from $p_\ell$ and is returned as the identified hypothesis (possibly wrong).

Now we already to introduce *chance-constrained ODT problem* (CC-ODT). Given an error budget $\delta > 0$, we aim to find the minimal cost decision tree whose error is within $\delta$. There are two natural ways to interpret "error", which both will be considered in Appendices C and D. In the first one, we require the error probability under *any* hypothesis to be lower than the given error budget. In the other one, we only require the *expected* error probability over all hypotheses to be within the budget. Intuitively, the second version allows for more flexibility since the errors under different hypotheses

may differ significantly, rendering the analysis more challenging since we do not know how the error budget is allocated to each hypothesis. We formalize these two versions below. Let $O$ be the random outcome returned by the tree.

**CC-ODT with PAC-Error.** An incomplete decision tree is $\delta$-*PAC-Valid* if, for any true hypothesis $h$, it returns $h$ with probability at least $1 - \delta$, formally,

$$\mathbb{P}_h(O \neq h) \leq \delta, \quad \forall h \in H.$$

**CC-ODT with Total-Error.** An incomplete decision tree is $\delta$-*Total-Valid* if, for the total error probability is at most $\delta$, formally,

$$\sum_{h \in H} \pi(h) \cdot \mathbb{P}_h(O \neq h) \leq \delta,$$

where $\pi$ is the prior distribution. The goal in both versions is to find an incomplete decision tree with minimal expected cost, subject to the corresponding error constraint.

For the proof of Proposition 2, consider the PAC-error version of CC-ODT. It turns out that this version of CC-ODT is indeed quite trivial (unlike the total-error version): below we show that under PAC-error, CC-ODT is almost equivalent to the ordinary ODT problem.

**Lemma 6.** Suppose $\delta \in (0, \frac{1}{2})$, and $\mathbb{T}$ is a $\delta$-PAC-valid decision tree. Then, $\mathbb{T}$ must also be 0-valid.

*Proof.* It suffices to show that there is no incomplete node in $\mathbb{T}$. For the sake of contradiction, assume $\mathbb{T}$ has an incomplete node $\ell$ with state $(A_\ell, p_\ell)$. By the definition of incomplete node, $|A_\ell| \geq 2$, so there is an $h \in A_\ell$ with $p_\ell(h) \leq \frac{1}{2}$. Now suppose $h$ is the true hypothesis. Since each hypothesis traces a unique path in any decision tree, regardless of whether or not it is incomplete, $h$ will reach node $\ell$ with probability 1. Then at $\ell$, the decision tree returns $h$ with probability $p_\ell(h) = 1 - \sum_{g \in A_\ell : g \neq h} p_\ell(g) \leq \frac{1}{2}$, and hence $\mathbb{P}_h[O \neq h] \geq \frac{1}{2}$, reaching a contradiction. $\square$

For the reader's convenience, we recall that an ASHT instance $\mathcal{I}$ is associated with an ODT instance $\mathcal{I}_{ODT}$, defined as follows. Each action corresponds to a test $T_a : H \to \Omega_a$ with $T_a(h) = \mu(h, a)$, where $\Omega_a = \{\mu(h, a) : h \in H\}$, and the cost $T_a$ is $c(a) = \lceil s(a)^{-1} \log(|H|/\delta) \rceil$. Denote $ODT_\delta^*$ the minimal cost of any $\delta$-PAC-valid decision tree for $\mathcal{I}_{ODT}$. Then we immediately obtain the following from the Lemma 6.

**Corollary 2.** If $\delta \in (0, \frac{1}{2})$, then $ODT_0^* = ODT_\delta^*$.

Now we are able to complete the proof of the main proposition.

**Proof of Proposition 2.** Given a $\delta$-PAC-error algorithm $\mathbb{A}$, we show how to construct a $\delta$-PAC-valid decision tree $\mathbb{T}$ as follows. View $\mathbb{A}$ as a decision tree (discretize the outcome space if it is continuous). Replace each action $a$ in $\mathbb{A}$ with the test $T_a$. Note that the cost of $T_a$ is $s(a)^{-1} \log(|H|/\delta) \leq s^{-1} \log(|H|/\delta)$. Therefore by Lemma 6,

$$ODT_0^* = ODT_\delta^* \leq c(\mathbb{T}) \leq s^{-1} \log \frac{|H|}{\delta} \cdot OPT_\delta^{FA}. \qquad \square$$

# D   Total Error Version

In the last section we defined the total-error version of the CC-ODT problem. The total error version of the ASHT problem can be defined analogously, so we do not repeat it here. We say an algorithm is said to be $\delta$-**total-error** if the total probability (averaged with respect to the prior $\pi$) of erroneously identified a wrong hypothesis is at most $\delta$. The following is our main result for the total-error version.

**Theorem 3.** Given an $s$-separated instance with uniform prior $\pi$ and any $\delta \in (0, 1/4)$, for both the partially and fully adaptive versions, there exist polynomial-time $\delta$-total-error algorithms with expected cost $O\left(s^{-1} \left(1 + |H|\delta^2\right) \log\left(|H|\delta^{-1}\right) \log |H|\right)$ times the optimum.

In particular, if $\delta \leq O(|H|^{-1/2})$, then the above is polylog-approximation for fixed $s$.

We will first prove Theorem 3 for the fully adaptive version, and then show how the same proof works for the partially adaptive version. Unlike the PAC-error version where CC-ODT is almost equivalent to ODT, in the total-error version their optima can differ by a $\Omega(|H|)$ factor. We construct a sequence of ODT instances $\mathcal{I}_n$, where $n \in \mathcal{Z}^+$, with $ODT_\delta^*(\mathcal{I}_n)/ODT_0^*(\mathcal{I}_n) = O(\frac{1}{n})$. Suppose there are $n + 2$ hypotheses $h_1, ..., h_n$ and $g, h$, with $\pi(g) = \pi(h) = 0.49$ and $\pi(h_i) = \frac{1}{50n}$ for $i = 1, ..., 50$. Each (binary) test partitions $[n + 2]$ into a singleton and its complement. Consider error budget $\delta = \frac{1}{4}$, then for each $n$ we have $ODT_\delta^*(\mathcal{I}_n) = 1$. In fact, we may simply perform a test to separate $g$ and $h$, and then return the one (out of $g$ and $h$) that is consistent with the outcome. The total error of this algorithm is $1/50 < \delta$. On the other hand, $ODT_0^*(\mathcal{I}_n) = n + 1$.

However, for uniform prior, this gap is bounded:

**Proposition 7.** Suppose the prior $\pi$ is uniform. Then, for any $\delta \in (0, \frac{1}{4})$, it holds
$$ODT_0^* \leq \left(1 + O(|H|\delta^2)\right) \cdot ODT_\delta^*.$$

To show the above, we need the following building block.

**Lemma 7.** Suppose the prior $\pi$ is uniform. Then, for any $\delta \in [0, \frac{1}{4})$, the total prior probability density on the incomplete nodes is bounded by $\sum_{\ell \text{ inc.}} \pi(A_\ell) \leq 2\delta$.

*Proof.* Let $\ell$ be an incomplete node with state $(A_\ell, p_\ell)$ and write $p = p_\ell$ for simplicity. Then, the error probability contributed by $\ell$ is
$$\sum_{h \in A_\ell} \pi(h) \cdot (1 - p(h)) = \sum_{h \in A_\ell} \pi(h) - \sum_{h \in A_\ell} \pi(h) \cdot p(h)$$
$$= \pi(A_\ell) - \frac{1}{n} \sum_{h \in A_\ell} p(h)$$
$$= \frac{|A_\ell|}{n} - \frac{1}{n} \geq \frac{1}{2}\pi(A_\ell),$$
where the last inequality follows since $|A_\ell| \geq 2$. By the definition of $\delta$-PAC-error, it follows that
$$\delta \geq \sum_{\ell \text{ inc.}} \sum_{h \in A_\ell} \pi(h) \cdot (1 - p(h)) \geq \frac{1}{2} \sum_{\ell \text{ inc.}} \pi(A_\ell),$$
i.e., $\sum_{\ell \text{ inc.}} \pi(A_\ell) \leq 2\delta$. $\square$

**Proof of Proposition 7.** It suffices to show how to convert a decision tree $\mathbb{T}$ with $\delta$-total-error to one with $0$-total-error, without increasing the cost by too much. Consider each incomplete node $A_\ell$ in $\mathbb{T}$. We will replace $A_\ell$ with a (small) decision tree that uniquely identifies a hypothesis in $A_\ell$. Consider any distinct hypotheses $g, h \in A_\ell$. Then by Assumption 2, there is an action $a \in A$ with $d(g, h; a) \geq s$. So if we select $T_a$, then by Hoeffding bound (Theorem 6), we have that with high probability at least one of $g$ and $h$ will be eliminated, and the number of alive hypotheses in $A_\ell$ reduces by at least 1. Thus, by repeating this procedure iteratively for at most $|A_\ell| - 1$ times, we can identify a unique hypothesis. Since each test $T_a$ corresponds to selecting $a$ for $c(a) = s(a)^{-1} \log(|H|/\delta) \leq s^{-1} \log(|H|/\delta)$ times in a row, this procedure increases the total cost by $\sum_{\ell \text{ inc.}} \pi(A_\ell) \cdot (|A_\ell| \cdot s^{-1} \log(|H|/\delta))$. Therefore,
$$ODT_0^* \leq ODT_\delta^* + \sum_{\ell \text{ inc.}} \pi(A_\ell)|A_\ell|s^{-1} \log \frac{|H|}{\delta}$$
$$= ODT_\delta^* + \sum_{\ell \text{ inc.}} \pi(A_\ell)|H|\pi(A_\ell) \cdot s^{-1} \log \frac{|H|}{\delta}$$
$$= ODT_\delta^* + O\left(s^{-1}|H| \log \frac{|H|}{\delta} \cdot \sum_{\ell \text{ inc.}} \pi(A_\ell)^2\right). \tag{6}$$

Since $\sum_{\ell \text{ inc.}} \pi(A_\ell) \leq 2\delta$ and each $\pi(A_\ell)$'s is non-negative, we have $\sum_{\ell \text{ inc.}} \pi(A_\ell)^2 \leq \left(\sum_{\ell \text{ inc.}} \pi(A_\ell)\right)^2 \leq 4\delta^2$. Further, by Pinsker's inequality, we have $ODT_\delta^* = \Omega(s^{-1} \log \frac{|H|}{\delta})$. Combining these two facts with Equation (6), we obtain $ODT_0^* \leq \left(1 + O(|H|\delta^2)\right) \cdot ODT_\delta^*$. $\square$

The following lemma can be proved using the same idea of the proof of Proposition 2.

**Algorithm 3 Partially Adaptive Algorithm in the COSMIC Experiment**

1: **Parameters**: Coverage saturation level $B > 0$ and maximum sequence length $\eta > 0$.
2: **Input**: ASHT instance $(H, A, \pi, \mu)$
3: **Output**: actions sequence $\sigma$
4: **Initialize**: $\sigma \leftarrow \emptyset$                                                       % Store the selected of actions
5: **for** $t = 1, 2, ..., \eta$ **do**                              % **Rank**: Compute a sequence of actions of length $\eta$
6:     $S \leftarrow \{\sigma(1), ..., \sigma(t-1)\}$.                                      % Actions selected so far
7:     **for** $a \in A$ **do**,                                                % Compute scores for each action

$$\text{Score}(a; S) \leftarrow \sum_{h:f_h^B(S)<1} \pi(h)\frac{f_h^B(S \cup \{a\}) - f_h^B(S)}{1 - f_h^B(S)}.$$

8:     **end for**
9:     $\sigma(t) \leftarrow \arg\max\{\text{Score}(a; S) : a \in A\}$.          % Select the greediest action and break ties
       according the heuristic described in Algorithm 4
10: **end for**
11: Let i be the largest index for which the an action appears the first time in sequence $\sigma$, then we
    return the sequence $(\sigma(1), ...., \sigma(i))$.

**Lemma 8.** $ODT_\delta^* \leq O\big(s^{-1}\log(|H|/\delta)\big)OPT_\delta^{FA}$.

Now we are ready to show Theorem 3.

$$\begin{aligned}
GRE &\leq O(\log|H|) \cdot ODT_0^* & \text{(Theorem 5)} \\
&\leq O\big((1 + O(|H|\delta^2))\log|H|\big) \cdot ODT_\delta^* & \text{(Lemma 7)} \\
&\leq O\big((1 + O(|H|\delta^2))s^{-1}\log^2\frac{|H|}{\delta}\log|H|\big) \cdot OPT_\delta^{FA}. & \text{(Lemma 8)}
\end{aligned}$$

The above proof can be adapted to the partially adaptive version straightforwardly as follows. Observing that partially adaptive algorithms can be viewed as a special case of the fully adaptive, we can define $ODT_{0,\text{PA}}^*$ and $ODT_{\delta,\text{PA}}^*$ (analogous to $ODT_0^*$ and $ODT_\delta^*$) for the partially adaptive version, as the optimal cost of any partially adaptive decision tree with 0 or $\delta$ error. By replacing $ODT_\delta^*$ and $ODT_0^*$ with $ODT_{0,\text{PA}}^*$ and $ODT_{\delta,\text{PA}}^*$, one may immediately verify that inequalities in Lemma 7 and 8 hold for the partially adaptive version. Furthermore, the first inequality above can be established for the partially adaptive version by replacing Theorem 5 with Theorem 4, hence completing the proof.

## E    Partially Adaptive Algorithm in Experiments

In our synthetic experiments, we implement Algorithm 1 described in Section 4 exactly, and set the boosting factor, $\alpha$, to be 1. In our real-world experiments, we consider a variant of our algorithm where the boosting intensity is now built-in in the algorithm, and breaking ties according to some heuristic. Algorithm 3 describes our modified algorithm. In particular, we consider the amount of boosting as a built-in feature of the algorithm. We first generate a sequence of actions of length $\eta$ for some large $\eta$ value (with replacement) and then truncate the sequence to the minimum length to include all unique actions that have appeared in the sequence. When all actions in sequence $\sigma$ has performed and we did not reach the target accuracy, then we repeat the entire sequence again. Our partially adaptive algorithm on the COSMIC data was generated by initializing $\eta$ to be 800. Across all accuracy levels, the maximum truncated sequence length is 97.

## F    Fully Adaptive Algorithm in Experiments

Similar to NJ's algorithm, we maintain a probability distribution, $\rho$, over the set of hypotheses to indicate the likelihood of each hypothesis being the true hypothesis $h^*$. A hypothesis is considered to be ruled out at each step if the probability of that hypothesis is below a threshold in $\rho$. Throughout our experiments, we set this threshold to be $\delta/|H|$. At each step, after an action is chosen with certain repetitions and observation(s) is (are) revealed, we update $\rho$ according to the realizations that we

observed. Thus, under this setup, a hypothesis that was considered to be ruled out in the previous steps (due to "bad luck") could potentially become alive again.

At each iteration, for each action $a \in A$ and $k \in \mathbf{N}$, we define $T_{a,k}$ to be the meta-test that repeats action $a$ for $k$ times consecutively, and we define its cost to be $c(T_{a,k}) = kc_a$. By Chernoff bound, with $k$ i.i.d. samples, we may construct a confidence interval of width $\sim k^{-1/2}$. This motivates us to rule out the following hypotheses when $T_{a,k}$ is performed. Let $\bar{\mu}$ be the observed mean outcome of these $k$ samples. We define the elimination set to be

$$E_{\bar{\mu}}(T_{a,k}) := \{h : |\mu(h,a) - \bar{\mu}| \geq Ck^{-1/2}\},$$

where C is set to be $1/2$ throughout our experiments. To define greedy, we need to formalize the notion of bang-per-buck. Suppose $H_{alive}$ is the current set of alive hypotheses. We define the score of a test as the number of alive hypotheses ruled out in the worst-case over all possible mean outcomes $\bar{\mu}$. Formally, the score of $T_{a,k}$ w.r.t mean outcome $\bar{\mu}$ is

$$\mathrm{Score}_{\bar{\mu}}(T_{a,k}) = \mathrm{Score}_{\bar{\mu}}(T_{a,k}; H_{alive}) = \frac{|E(T_{a,k}; \bar{\mu}) \cap H_{alive}|}{c(T_{a,k})},$$

and define its worst-case score to be

$$\mathrm{Score}(T_{a,k}) = \min\{\mathrm{Score}_{\bar{\mu}}(T_{a,k}) : \bar{\mu} \in \{0, 1/k, ..., 1\}\}.$$

Our greedy policy simply selects the test $T$ with the highest score, formally, select

$$T_{a,k} = \arg\max\{\mathrm{Score}(T) : k \leq k_{\max}, a \in A\}.$$

In the synthetic experiments, we set $k_{\max} = 5$. In the real-world experiments, we consider the cases where $k \in \{15, 20, 25, 30\}$ (with $k_{\max} = 30$).

If several actions have the same greedy score, then we choose the action $a^*$ whose sum of the KL divergence of pairs of $\mu(h, a^*)$ is the largest, and breaking ties arbitrarily.

If no action can further distinguish any hypotheses in the alive set, then we set the boosting factor to be 1 and use the above heuristic to choose the action to perform. The algorithm is formally stated in Algorithm 4.

**Algorithm 4 Adaptive experiments: FA($k_{\max}, \delta$)**

---

1: **Parameters**: maximum boosting factor $k_{\max} > 0$ and convergence threshold $\delta > 0$
2: **Input**: ASHT instance $(H, A, \pi, \mu)$, current posterior about the true hypothesis vector $\rho$
3: **Output**: the test $T_{a,k}$ to perform in the next iteration
4: Let $H_{\mathrm{alive}}$ be the set of hypotheses $i$ whose posterior probability $\rho_i$ is above $\delta/|H|$.
5: **for** $k = 1, 2, ..., k_{\max}$ **do**
6:     For each $a \in \widetilde{A}$ define:

$$\mathrm{Score}_{\bar{\mu}}(T_{a,k}) = \mathrm{Score}_{\bar{\mu}}(T_{a,k}; H_{\mathrm{alive}}) = \frac{|E(T_{a,k}; \bar{\mu}) \cap H_{\mathrm{alive}}|}{c(T_{a,k})},$$

    where $E_{\bar{\mu}}(T_{a,k}) := \{h : |\mu(h, a) - \bar{\mu}| \geq Ck^{-1/2}\}$, and $c(T_{a,k}) = kc_a$. We define the worst-case score of a test to be:

$$\mathrm{Score}(T_{a,k}) = \min\{\mathrm{Score}_{\bar{\mu}}(T_{a,k}) : \bar{\mu} \in \{0, 1/k, ..., 1\}\}.$$

7: **end for**
8: Compute greediest action

$$G = \arg\max\{\mathrm{Score}(T) : k \leq k_{max}, a \in A\}.$$

9: **if** the Score of each test in $G$ equals to 0, i.e, no test can further distinguish between the alive hypotheses under $k_{\max}$ **then**
10:     we choose the action $a^*$ such that $a^* = \arg\max \sum_{h,g \in H_{\mathrm{alive}}} \mathrm{KL}(\mu(h, a), \mu(g, a))$, breaking ties randomly, and return $k = 1$.
11: **else**
12:     if $G$ is a singleton, then we return $G$. Else, we choose the action $a^*$ such that $a^* = \arg\max_G \sum_{h,g \in H_{\mathrm{alive}}} \mathrm{KL}(\mu(h, a), \mu(g, a))$, and breaking ties randomly.
13: **end if**

---