# OpenReview forum: "Greedy Approximation Algorithms for Active Sequential Hypothesis Testing"
_NeurIPS.cc/2021/Conference — NeurIPS 2021 Poster_

### Official Review · Reviewer_MCYe · 2021-07-16

**Rating:** 7
**Confidence:** 3

**Summary:**

The authors consider the problem "active sequential hypothesis testing", where the learner finds the true hypothesis by observing the noisy outcomes of actions. In particular, they study the ASHT under two types of adaptivity: partial and full. For both these settings, they propose greedy algorithms that run efficiently and provide an approximate solution.

**Limitations And Societal Impact:**

The authors do a sufficient job of addressing the limitations of their work.

**Main Review:**

Things that I like about the paper:
1) The problem considered in this paper is generic and captures a number of well-known problems; I find this generalization interesting.
2) The paper is well written and the authors do a good job in placing their contribution with respect to the related work.
3) The paper is a nice mixture of both theory and practice. The experiments section is well done and I enjoyed reading the performance of the proposed algorithm on real data.


Few concerns/suggestions:
1) Why is assumption 1 needed in general? Some intuition on why is it hard to overcome assumption 1 would be helpful.
2) One of the significances of the approximation ratios is that they are independent of the number of actions and in applications where the action space is massive the presented algorithms are useful. However, note that the algorithm runs in O(|A|) time (hiding other dependencies)  and therefore might be impractical in the aforementioned applications. I am aware that removing the dependency of O(|A|) in run time might not be possible but it would be interesting to replace the |A| term in the running time with some other structural property of A. For instance, with the term $O_{A}$, where $O_{A}$ is the time to perform a particular action on A and then mention that $O_{A}<=A$ in the worst case and provide examples/applications where $O_{A}$ would be much smaller.
3) Talk about what is the value of $s$ (Definition 1) in typical applications. I see that the first few sentences in the experiments section handle this concern, it would be nice to add some of it when $s$-separation is defined.
4) If possible, mention approximation ratios (instead of just No and Yes) in Table 1.
5) Consider other benchmarks in the experiments or explain why you don't include them.
6) When mentioning theory results, talk about the challenges faced in achieving these results. For instance, the difficulty in the analysis and how you improve previous works to overcome these difficulties. There are few sentences in the introduction that handle this concern, but I would like to see more sentences in the early stages of the paper.

**Time Spent Reviewing:**

8

---

> ### Author Response · Authors · 2021-08-10
> **Detailed Comments to Reviewer MCYe**
>
> Thank you for your thoughtful and constructive comments.  Please see below for our detailed response.
>
> (1) "Why is assumption 1 needed in general? Some intuition on why is it hard to overcome assumption 1 would be helpful."
>
> **Response.**  At a high level, Assumption 1 is needed for relating the sub-gaussian norm to the KL-divergence, in the partially adaptive version (and we have added this point in Lines 87-88 of our revised paper). In fact, on the one hand, the stopping time depends on the total KL-divergence collected by the actions selected so far.
> On the other hand, to prove that the stopping time is sufficient for achieving $\delta$-error, we apply Hoeffding's inequality to bound the error probability in terms of the sum of *squared* sub-Gaussian norms. To complete the proof, we need Assumption 1 to relate these two quantities. It is straightforward to obtain a weaker but more general result by removing this assumption, but for the sake of readability we decided not to pursue this route.
>
> (2) "One of the significances of the approximation ratios is that they are independent of the number of actions and in applications where the action space is massive the presented algorithms are useful. However, note that the algorithm runs in O($|A|$) time (hiding other dependencies) and therefore might be impractical in the aforementioned applications. I am aware that removing the dependency of O($|A|$) in run time might not be possible but it would be interesting to replace the $|A|$ term in the running time with some other structural property of A. For instance, with the term, where  is the time to perform a particular action on A and then mention that  in the worst case and provide examples/applications where  would be much smaller"
>
> **Response.** This is a good point and we agree that the running time may be improved at the cost of the approximation factor, assuming a special structure in the input. For both partially and fully adaptive versions, if we select a c-approximate greediest action (i.e. whose greedy score is $\geq \frac 1c$ times the maximum) instead of the exact greediest test, the approximation factor will inflate by c times. In particular, if the input -- which is essentially an $|H| \times |A|$ table -- is, say, randomly generated, then it is possible to improve the running time by sampling a subset of tests and argue that the action computed is nearly-greediest. Exploring special types of input to the ASHT or ODT problem may be intriguing directions for future work.
>
> Finally, it is worth noting that in our experiments we implemented the greedy algorithm by parallelism. This was feasible since Score$(a; S)$ (in line 6 of Algorithm 1) can be calculated independently for each action $a$.
>
> (3) "Talk about what is the value of  (Definition 1) in typical applications. I see that the first few sentences in the experiments section handle this concern, it would be nice to add some of it when s-separation is defined."
>
> **Response.** Thank you for the suggestion. We have added the following sentences after Definition 1:
> *"Note that in real-world applications, the parameter $s$ could be arbitrarily small, and we introduce the notion of s-separability for the purpose of our proofs. We will show in Section 6 how our algorithms can easily be modified to handle small $s$ values."*
>
> (4) "If possible, mention approximation ratios (instead of just No and Yes) in Table 1."
>
> **Response.** Thank you for the suggestion. The approximation ratios for both row 3,4 are $O(\log |H|)$. We are however unable to summarize our approximation ratios in the cell in row 6 since we considered both partially and fully adaptive versions.
>
> (5) "Consider other benchmarks in the experiments or explain why you don't include them."
>
> **Response.** Thank you for the suggestion. We did not include other benchmarks since there are not many other computationally tractable and directly comparable benchmarks apart from Naghshvar and Javidi [34]. We added the following at the beginning of Section 6:
>
> *"To the best of our knowledge, the only known polynomial-time policy for ASHT is proposed by [34], which will serve as the benchmark for our experimental section."*
>
> (6) "When mentioning theory results, talk about the challenges faced in achieving these results. For instance, the difficulty in the analysis and how you improve previous works to overcome these difficulties. There are few sentences in the introduction that handle this concern, but I would like to see more sentences in the early stages of the paper."
>
> **Response.** There is a paragraph (Lines 185-197) that explains the challenges in obtaining provable guarantees for the partially adaptive version. We agree that it is better to move it to the introduction. The analysis of the fully adaptive version is somewhat more straightforward so we did not explain its difficulty with a separate paragraph, but we will improve this in our revised paper.

---

### Official Review · Reviewer_qxXc · 2021-07-16

**Rating:** 6
**Confidence:** 3

**Summary:**

This work presents approximation algorithms for the problem of active sequential hypotheses testing. The proposed algorithms leverage a connection with submodular function ranking that is novel and interesting. Both "fully adaptive" and "partially adaptive" algorithms are given, with different features and guarantee. The experimental evaluation of (minor variants) of the proposed algorithms shows that they seem to outperform existing approaches.

**Limitations And Societal Impact:**

The Authors do a pretty good job at assessing the *algorithmic* limitations of the work.

This Reviewer does not see any potential negative societal impact that is *specific* to this work.

**Main Review:**

## Comments after response

I thank the Authors for their response. I read it carefully, as I did the comments from other Reviewers and the responses to them.
I'm keeping my score as it is, but I think the paper should be accepted, even if it just marginally above the threshold.

## Original Review

This work is the first proposal of algorithms offering quality guarantees for the studied problem. While the guarantees are not particularly strong, and the computational complexity ($O(|A||H|)$) is quite high, the work seems solid and it opens a new direction. It also shows interesting connection with submodular function ranking and the existing approximation algorithms for it. The use of "boosting" (repeating an action multiple times) is also a nice touch, although it seems to be the point where theory and practice diverge the most. The experimental evaluation seems quite convincing, with the exception of an issue in the selection of tests to consider (detailed below).

The paper is generally well written. There is clearly an effort in carefully explaining the connections with other problems and the general intuition about the proposed approach.

## Other comments

Lines 77-83: this part feels a bit too informal, given that it describes very important quantities such as the $\delta$-PC-error and what is meant by "sufficiently high" confidence. While these concepts are explained in words, a mathematical expression may ground them better.

Line 185: when the reader gets here, they have no idea of what is meant by "boosting".

Line 347: why can one remove all these duplicated tests? They may share the same outcome distribution for all 8 cancer types on this data, but not in general, so it is unclear whether one can really filter them out. But maybe it is allowed (i.e., no guarantees whatsoever are lost). If so, it should be made quite clear

377: "approximationfactors" -> "approximation factors"

**Needs Ethics Review:**

Yes

**Time Spent Reviewing:**

2.5

---

> ### Author Response · Authors · 2021-08-10
> **Detailed response to qxXc**
>
> Thank you for your thoughtful and constructive comments.  Please see below for our detailed response.
>
> (i) **Lines 77-83:**  "This part feels a bit too informal, given that it describes very important quantities such as the -PC-error and what is meant by "sufficiently high" confidence. While these concepts are explained in words, a mathematical expression may ground them better."
>
> **Response.** Thank you for the suggestion. To formally define the error of an algorithm, we need to first formally define an "algorithm" (see Lines 104-111). We propose to add the following right after line 117 to clarify the meaning of the error:
>
> *"Let $O$ be the random outcome of an algorithm, then the PAC-error is defined to be
> $\sum_{h\in H} \pi_h \cdot P_h[O\neq h]."$*
>
> (ii) **Line 185:** "when the reader gets here, they have no idea of what is meant by "boosting"."
>
> **Response.** Thank you for the suggestion, the word "boost" simply means repeating an action for a sufficient number of times so that by concentration bounds, the outcome is almost deterministic. Our revised sentence now reads as *"...'boost' (or repeat) each action for sufficient number of times..."}*.
>
> (iii) **Line 347:** "Why can one remove all these duplicated tests? They may share the same outcome distribution for all 8 cancer types on this data, but not in general, so it is unclear whether one can really filter them out. But maybe it is allowed (i.e., no guarantees whatsoever are lost). If so, it should be made quite clear."
>
> **Response.** This is because those tests are equivalent when restricted to these 8 cancer types, and we allow a test to be selected for multiple times in our experiment. We agree that a pair of equivalent tests on those 8 cancer types may no longer be equivalent on a larger set of cancer types. We propose to add the following right after line 347 to clarify:
>
> *"Note that the duplicate tests can be removed here since they are equivalent if restricted on these 8 cancer types."*
>
> (iv) **Line 377:** " 'approximationfactors' $\rightarrow$ 'approximation factors'.''
>
> **Response.** Thank you, we have fixed the typo.

---

### Official Review · Reviewer_ACVf · 2021-07-17

**Rating:** 6
**Confidence:** 5

**Summary:**

The problem considered in the paper regards the identification of the "correct one" (or true) hypothesis, chosen from a set of candidate hypotheses according to some given prior distribution. The identification works by applying tests/actions taken from a given set. The test returns a probabilistic result, according to a distribution that depends on the test and the hypothesis. The problem is to choose the sequence of tests so that from the results the correct hypothesis can be identified with probability bounded by some given parameter delta.
The model is a variant of the more classical combinatorial identification problems, where the outcome of the tests is a deterministic value. In fact, as, admittedly, the authors pointed out the problem remains inherently combinatorial and the solution provided here are tightly connected to the known approximation solutions in the non-statistical variants. The basic strategy is to apply the latter and repeat each test enough time to identify a signle outcome that have statistical significance. Then, proceed according to the deterministic strategy. Standard and some new statistical analytical tools are effectively employed to guarantee that from the random outcome a deterministic result is chosen that guarantess the separation (in the fully adaptive case) or coverage (in the nonadaptive case) leading to the overall cost bound.

**Limitations And Societal Impact:**

The work does not pose potential negative societal impact. The authors addressed the main limitations in the final section.

**Main Review:**

Originality: This is an interesting generalization of some well-studied problems (e.g. Active learning and Optimal Decision Trees) that are relevant to the NeurIPS community.  The ASHT variant considered here is well motivated by applications like the cancer/genomics-data biological scenario considered in the experiment section. The main contribution claimed by the authors is mainly of theoretical nature and consists in the first approximation bounds for this version of the problem, in both the adaptive and semi-adaptive case.

Strength: The authors present approximation guarantees for the Active Sequential Hypothesis Testing Problem which depend logarithmically in |H| and the inverse of the separability parameter of the instance, a parameter that describes the minimum positive distance between distinct outcome distributions associated with some action.

Weakness: There are some important issues with one of the main claims that should be clarified by the authors. Theorem 5 (which is taken from [Chakaravarthy et al. ICALP2009] does not appear to be used in the correct way in the present paper for a couple of reasons: (i) the ICALP result is about the variant of the ODT problem where tests have no costs, while in this paper costs appear to be necessary to model the number of times a test is used to achieve statistical significance; (ii) the authors claim that the result is for arbitrary distribution, but the part that leverage on the ICALP result cannot be extended without referring to different model, which might necessitate a different connection between  the ASHT and the "average case - with variable test-costs" variant of the ODT problem.

Clarity: In general the paper is clearly written. The only vague parts are those that concern the issue above. I think the authors should give further details to clarify how, besided the limitation of the ICALP paper, they manage to close the gaps between the model of ODT they need to support the simplified analysis of ASHT that is only based on [Chakaravarthy et al. ICALP2009] result.

Overall, I consider the problem and the claimed result interesting. My evaluation is set just below acceptance because of the above doubts, that I hope the authors will be able to clarify.

**Time Spent Reviewing:**

6

---

> ### Author Response · Authors · 2021-08-10
> **Correctness and Generalizability of Theorem 5**
>
>  Thank you for your thoughtful and constructive comments. Please see below for our response to your comments.
>
> **Response to Weakness point (i):**
>
> We apologize for not explaining the proof clearly enough.
> We were in fact aware that the Chakaravarthy et al 09 analysis holds only for uniform test costs, and that's why the greedy score in Algo 2 is not divided by $c(\hat a)$.
> Theorem 5 is indeed *correct* if we slightly revise the reduction to ODT as follows.
>
> For the sake of analysis, in addition to $c(a)$ as defined in line 255, we consider a "fake" cost $c' := \lceil s^{-1} \log (|H|/\delta) \rceil$, which does not depend on $a$.
> The definition of the ODT instance $I_{ODT}$ remains the same except that each test has **uniform** cost $c'$ (as opposed to $c(a)$). Let $c(T)$ and $c'(T)$ be the costs of the greedy tree $T$ returned by Algo 2 under $c$ and $c'$ respectively.
> Then by Chakravorthy et al '09, we have
> $$c'(T) \leq O(\log |H|)\cdot ODT^*.$$
>
> Note that $c'\leq c(a)= \lceil s(a)^{-1} \log (|H|/\delta) \rceil$ for each $a$ since, by definition, the separation parameter $s$ is no larger than $s(a)$.
> Hence the (real) cost of Algo 2 satisfies
> $$GRE \equiv c(T)\leq c'(T).$$
> One can easily verify that Proposition 2 still holds, since its proof only involves boosting each action in $OPT^\delta_{PA}$ for the same number of times, and is not affected by the change made above. Thus,
> $$ODT^* \leq s^{-1}\log (|H|/\delta)\cdot OPT_{PA}^\delta.$$
> Combining, we obtain
> $$GRE\leq c'(T) \leq O(\log |H|)\cdot  ODT^* \leq O\left(s^{-1}\log |H| \cdot\log (|H|/\delta)\right) \cdot OPT_{PA}^\delta,$$
> and the proof follows.
>
> **Response to Weakness point (ii):**
>
> In Theorem 5 (line 248), we only claimed the result for uniform prior. Later in line 274, we remarked that "the analysis can be easily extended to general priors by reduction to the Adaptive Submodular Ranking (ASR) problem (Navidi, Kambadur and Nagarjan, IPCO'17)."
>
> In fact, since ASR includes ODT as a special case, one may easily verify that the main theorem in the aforementioned paper implies that a (slightly different) greedy algorithm achieves $O(\log |H|)$-approximation for the ODT problem with *general* prior, test costs, and an *arbitrary* number of branches in each test.
> Thus for general prior, the same analysis goes through if we first reduce ASHT to ASR, and then replace the greedy step (Step 4 in Alg 2) with the greedy criterion for ASR.
> We decided not to state and prove the general prior version in order to highlight the central idea and keep the main body simple.
>
> We apologize for being too sketchy in the remark in Lines 274-276. If needed, we are able to present a formal proof in the appendices.
>
> **Response to Clarity:**
>
> Thank you for your positive comments. We have addressed the clarity issue above.

---

### Review · Ethics_Reviewer_RdxE · 2021-08-11

**Recommendation:** N/A

**Ethics Review:**

Paper does not raise Ethical issues.

---

### Review · Ethics_Reviewer_1CRT · 2021-08-12

**Recommendation:**

The authors can add a discussion of broader impacts to the Conclusion section without substantive changes to the rest of the paper.

**Ethics Review:**

This reader and the other reviewers have not identified any specific ethical concerns to be addressed. The paper does not include a discussion of broader impacts, potential harms, or possible malicious use. This could be added to the Conclusions section without affecting the rest of the paper.

---

### Decision · Program_Chairs · 2021-09-27

**Decision:**

Accept (Poster)

**Comment:**

The paper considers the problem of active sequential hypothesis testing, where a sequence of actions are adaptively chosen to find the best hypothesis without testing all of them. The paper considers the important novel setting when the test returns a result that is stochastic, and proposes a greedy approach by linking the problem to submodular function ranking. Theoretical analysis and empirical results on the COSMIC cancer data are presented.

Three expert reviewers carefully considered the strengths and weaknesses of the paper. All reviewers agree that the problem setting is important, and the link to submodularity and the resulting greedy algorithm is interesting. They identified several issues with clarity and precision in the paper, which the authors tried to address during the rebuttal period, which the reviewers appreciated (and changed their scores). Post rebuttal, the reviewers had a brief discussion and agreed that the paper is a good one, and would be a valuable contribution to the NeurIPS program. The authors are strongly advised to take the reviewer feedback into account in the final version. I am pleased to be able to recommend this paper to be accepted to NeurIPS. Congratulations!